# SegPVSG: Panoptic Video Scene Graph Generation via Temporal Focusing and Generative Augmentation

**Yikai Li** [* 1]  **Quhui Ke** [* 1]  **Jinglin Liang** [1]  **Zhiyuan Zhang** [1]  **Zhidi Lin** [2]  **Shuangping Huang** [1 3]

## Abstract

Panoptic Video Scene Graph Generation (PVSG) aims to identify relations between pixel-level entities in a video, serving as a novel paradigm for structured video parsing. However, this task faces two key challenges. First, the interactions between entities are temporally fragmented and sparse, meaning videos are dominated by irrelevant content with limited salient information. Second, the distribution of relations exhibits a significant long-tailed pattern, making models struggle to perform well on tail categories with insufficient data. To address these issues, we propose SegPVSG, an innovative, temporal-segment-aware PVSG framework consisting of two key components: TempFocusNet (TFN) and Relation-centric Generative Video Augmentation (RGVA) module. TFN is a localization-then-recognition network that improves PVSG performance by explicitly localizing and focusing on salient segments before relation recognition. Meanwhile, RGVA is a novel augmentation module that generates realistic, context-consistent video segments for rare relations and coherently inserts them into original videos. Our method outperforms prior methods by +3.53 mR@20 and +5.9 mR@50, demonstrating its effectiveness. The code is available at https://github.com/ticatt/SegPVSG.

## 1. Introduction

Panoptic Video Scene Graph Generation (PVSG) is a structured video parsing task that converts raw videos into dynamic scene graphs by jointly modeling pixel-level entity trajectories and their temporal relations (Yang et al., 2023b).

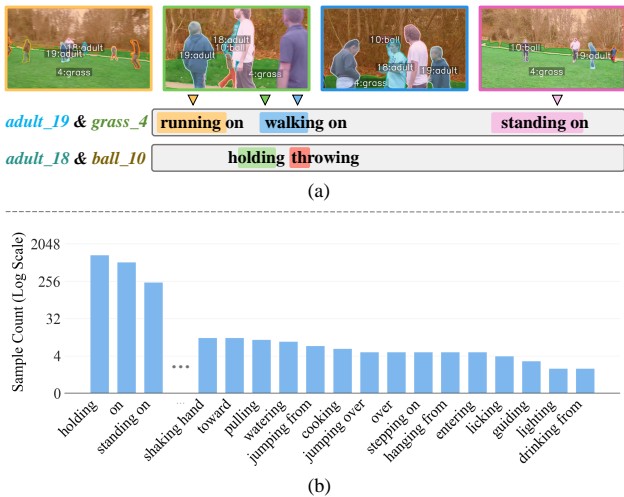

*Figure 1.* Critical challenges in PVSG. **(a) Temporal Sparsity:** Relations only occur in specific, often short, temporal segments of a video. **(b) Long-tailed Distribution:** The distribution of relation categories in PVSG datasets exhibits a severe long-tailed pattern.

In such graphs, nodes correspond to entity trajectories, and edges encode the temporal relations between entities. PVSG supports diverse downstream applications, including visual reasoning (Chu et al., 2025; Qiu et al., 2025), video question answering (Li et al., 2022b; Min et al., 2024), and dynamic environment guidance (Lan et al., 2025; Huang et al., 2026).

However, the PVSG task faces two critical challenges, as shown in Figure 1. First, relations in videos typically occur only in a few segments, leaving most frames dominated by background or unrelated actions that obscure salient moments. Second, relation categories exhibit a pronounced long-tailed pattern. For tail relations, scarce training data provides insufficient supervision and limited variation in the occurrence patterns of these relations, which limits model generalization.

Despite progress in prior work, fundamental challenges remain unresolved. For example, to extract meaningful temporal structures from noisy, background-dominated video sequences, existing methods employ full-sequence self-attention (Yang et al., 2023a;b; Nguyen et al., 2025b). However, these methods often suffer from attention drift, wherein visually salient yet semantically irrelevant background mo-

[*]Equal contribution  [1]South China University of Technology, Guangzhou, China [2]The University of Hong Kong, Hong Kong SAR, China [3]Pazhou Laboratory, Guangzhou, China. Correspondence to: Shuangping Huang <eehsp@scut.edu.cn>.

*Proceedings of the 43$^{rd}$ International Conference on Machine Learning*, Seoul, South Korea. PMLR 306, 2026. Copyright 2026 by the author(s).

tions dominate the learned representations. Alternative approaches leverage local information through sliding windows or heuristics to segment clips (Liu et al., 2020; Chen et al., 2021; Wei et al., 2024), but such hard partitioning restricts the receptive field and fragments the continuous temporal context. In parallel, to overcome the issue of the long-tailed pattern exhibited in relations, existing methods mainly perform calibration and redistribution within the original data and feature spaces. For instance, Peddi et al. (2025); Xu et al. (2022) apply gradient calibration to suppress head-category overfitting, while Nag et al. (2023); Li et al. (2025) perform feature transfer and enhancement to alleviate the underfitting of tail categories. Nevertheless, the issue of the scarce training data persists in these methods, as they essentially do not introduce any additional training instances for rare relations.

To address these limitations, we propose SegPVSG, a temporal-segment-aware framework. It comprises two key components: TempFocusNet (TFN), a localization-then-recognition paradigm for modeling video relations, and Relation-centric Generative Video Augmentation (RGVA), a data enrichment pipeline for rare relations. Specifically, TFN utilizes learnable queries to interact with global entity features, efficiently searching for potential relation intervals. These intervals are then modeled as Gaussian temporal masks, serving as soft attention constraints that guide the recognition process, allowing the model to focus on salient segments while preserving relevant global context. Furthermore, RGVA synthesizes self-contained interaction segments for rare relations by leveraging a Multimodal Large Language Model (MLLM) together with an image-to-video model, and then coherently inserts these segments into the original videos. Unlike generating video clips in isolation, this insertion strategy preserves the continuous temporal context of the original video, which is essential for learning interval-level temporal localization in PVSG. By enriching rare relations with diverse appearance and motion patterns, RGVA alleviates generalization issues for tail relations.

Briefly, our key contributions are summarized as follows:

- We propose SegPVSG, a temporal-segment-aware framework with TFN (§ 3.1), which employs Gaussian temporal masks in a localization-then-recognition paradigm to effectively model relations.

- We introduce RGVA (§ 3.2), which synthesizes interaction segments for rare relations to overcome scarce training data and enhance model generalization.

- Comprehensive experiments demonstrate that our approach achieves a significant improvement, with gains of +3.53/+5.90/+7.70 in mR@20/50/100 (§ 4.2).

## 2. Related Work

### 2.1. Panoptic Video Scene Graph Generation

Video Scene Graph Generation (VidSGG) generalizes image SGG by modeling dynamic relations over time (Chang et al., 2021). Existing methods fall into frame-level or video-level paradigms. Frame-level approaches (Nguyen et al., 2024; 2025a; Chen et al., 2025a; Wang et al., 2025) generate per-frame graphs, but typically model dynamics implicitly, providing limited structure for relation onset, duration, and termination. Conversely, video-level approaches (Jiang et al., 2024; Gao et al., 2022; Woo et al., 2025) associate entities across frames to infer relations over longer extents, improving temporal consistency. However, they commonly rely on aggregating fixed-length clips (Shang et al., 2017; Liu et al., 2020) or applying heuristic post-processing (e.g., watershed-style procedures) on interaction scores (Chen et al., 2021) to estimate relation duration. While effective, these pipelines often lack explicit end-to-end learning for precise relation boundaries.

Building on the video-level paradigm, PVSG introduces pixel-level supervision for fine-grained understanding (Yang et al., 2023b). Recent works have advanced this field through diverse paradigms: UNO (Le et al., 2026) introduces a one-stage architecture that jointly optimizes entity segmentation and relation prediction, while Click2Graph (Ruschel et al., 2025) explores interactive PVSG through user prompting. Others focus on enriching feature representations, such as utilizing motion cues (Nguyen et al., 2025b). Despite these architectural and interactive advances, learning relation intervals with explicit, end-to-end temporal modeling remains underexplored.

### 2.2. Long-tailed Learning in VidSGG

Severe class imbalance in VidSGG datasets often leads to biased predictions toward head relations. Early attempts mainly relied on loss re-weighting schemes (Yang et al., 2023b), while recent methods emphasize more structured debiasing within the learning pipeline. At the optimization level, IMPARTAIL (Peddi et al., 2025) mitigates head-class dominance via curriculum-guided loss masking that progressively suppresses their gradients. MVSGG (Xu et al., 2022) addresses spatio-temporal conditional biases with a meta-learning framework that simulates distribution shifts between support and query sets. At the feature level, TEMPURA (Nag et al., 2023) alleviates tail-class underfitting using a memory diffusion unit to transfer feature prototypes from data-rich to data-poor categories, and VISA (Li et al., 2025) calibrates visual and semantic biases through memory-enhanced temporal integration and hierarchical feature extraction. Despite their effectiveness, these approaches primarily redistribute supervision or calibrate representations over observed instances (Peng et al., 2025a;b), without

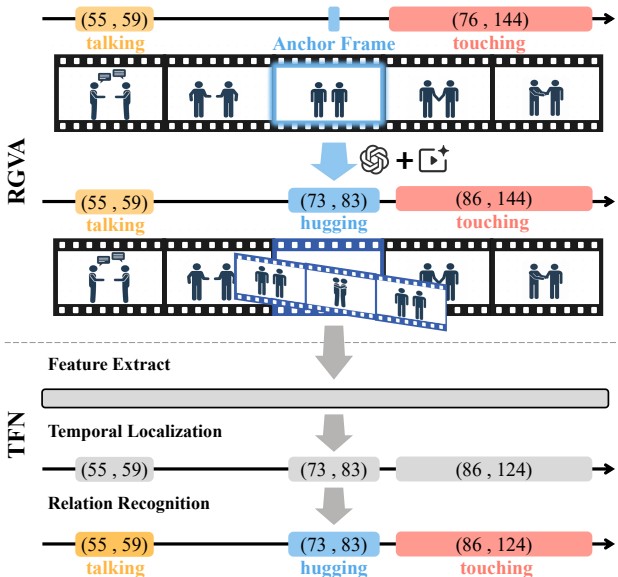

*Figure 2.* Overview of the SegPVSG framework. It consists of RGVA for enriching rare relations with self-contained interaction segments and TFN for interval-aware relation modeling.

introducing additional visual-temporal evidence. As a result, rare relations remain impoverished in appearance and motion diversity. In contrast, our method adopts a generative strategy to synthesize new short, self-contained interaction segments, thereby enriching the training data and enhancing the model's generalization performance.

## 3. Methodology: SegPVSG Framework

**Task Definition.** Given a video clip $V \in \mathbb{R}^{T \times H \times W \times 3}$ consisting of $T$ frames of spatial resolution $H \times W$, the PVSG task aims to construct a dynamic scene graph $G = (M, O, R)$ that captures entities and their spatio-temporal interactions. Formally, this is expressed as:

$$P(G \mid V) = P(M, O, R \mid V). \tag{1}$$

Specifically, $G$ comprises $N$ entities and $L$ temporal relation instances, where $N$ denotes the number of tracked entities in the video and $L$ denotes the number of subject-object relation instances. We represent the entity mask tubes as $M = \{m_i\}_{i=1}^N$, the corresponding entity category labels as $O = \{o_i\}_{i=1}^N$, and the temporal relations as $R = \{r_l\}_{l=1}^L$. Each entity $i$ is defined by a mask tube $m_i \in \{0, 1\}^{T \times H \times W}$ and a label $o_i \in \mathbb{C}^O$. Each relation instance $r_l$ encodes a predicate label in $\mathbb{C}^R$ between a subject-object pair over a specific temporal interval. Here, the sets $\mathbb{C}^O$ and $\mathbb{C}^R$ correspond to the object and predicate classes, respectively.

**Overview.** In this paper, we propose SegPVSG, a framework specifically tailored for the PVSG task, which comprises two key components: TFN and RGVA. As depicted in

Figure 2, TFN localizes and models relation-relevant temporal segments to improve interval-level relation recognition. RGVA synthesizes relation-centric augmented videos to enhance training data diversity, thereby specifically addressing the data scarcity issue for rare relations. Details of TFN and RGVA are presented in § 3.1 and § 3.2, respectively.

### 3.1. Temporal Relation Modeling

As illustrated in Figure 3, TFN adopts a localization-then-recognition paradigm to localize sparse interaction spans and suppress irrelevant temporal context. Given an input video, we first construct entity feature tubes containing visual and semantic cues via the *Entity Feature Extractor*. To handle dynamic interactions, the *Relation Interval Locator* employs learnable temporal queries to regress candidate intervals, which are uniquely modeled as Gaussian masks to capture the temporal structure. Finally, these masks serve as soft structural constraints in the *Temporal Focus Classifier*, guiding the attention mechanism to concentrate discriminative reasoning on relevant segments while filtering out irrelevant context. The remainder of this section details these components.

#### 3.1.1. ENTITY FEATURE EXTRACTOR

The Entity Feature Extractor aims to construct temporally consistent representations for all entities in a video. Given a frame, we employ Mask2Former (Cheng et al., 2022) to generate entity masks and category predictions. For each detected entity, we then perform mask pooling over the backbone feature map to extract a precise visual representation $f_v \in \mathbb{R}^D$, and map the predicted category index to a learnable semantic embedding $f_s \in \mathbb{R}^D$, where $D$ denotes the feature dimension. These visual and semantic cues are fused via a gated MLP to form the entity representation.

Following existing practices (Wang et al., 2021; Li et al., 2022a), we then associate entities across time to form entity feature tubes, denoted as $\{F_i\}_{i=1}^N$ (see § 4.1 for implementation details). Here $N$ is the number of tracked entities and each $F_i \in \mathbb{R}^{T \times D}$ represents the feature sequence of entity $i$ over $T$ frames. To model interactions, we construct the pairwise feature tube $F_{ij} = [F_i, F_j] \in \mathbb{R}^{T \times 2D}$ for subject $i$ and object $j$ via concatenation. During inference, we employ a pairing component to efficiently select candidate entity pairs based on pairwise similarity (Yang et al., 2023b).

#### 3.1.2. RELATION INTERVAL LOCATOR

We design the Temporal Query Decoder to generate candidate intervals. Then, each interval is encoded as a Gaussian mask to represent its internal temporal structure. These masks will guide the classifier (see § 3.1.3) to focus on the most salient frames, enabling precise relation recognition.

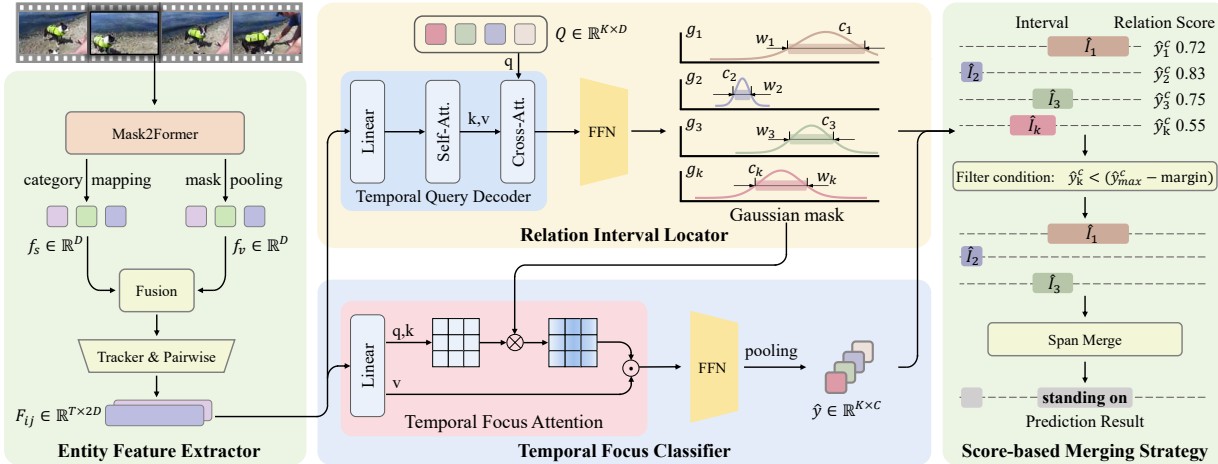

*Figure 3.* Overview of TFN. We first track entities to form candidate pairs with visual-semantic features. For each pair, temporal queries localize relation intervals via Gaussian masks, which guide Temporal Focus Attention to aggregate features for classification. Finally, a score-based strategy merges high-confidence intervals into final predictions.

**Temporal Query Decoder.** Given a selected entity pair $F_{ij} \in \mathbb{R}^{T \times 2D}$ (which we will omit subscripts $(i, j)$ for brevity hereafter), we first project it to a $D$-dimensional space, after which its temporal context is encoded via a self-attention layer, $SA(\cdot)$, yielding the representation $\bar{F} \in \mathbb{R}^{T \times D}$. Then, $K$ learnable temporal queries $Q \in \mathbb{R}^{K \times D}$ interact with $\bar{F}$ via a cross-attention layer, $CA(\cdot)$, to generate interaction features $H \in \mathbb{R}^{K \times D}$. This design enables queries to capture global interaction context and adaptively localize diverse dynamic relation intervals.

Having obtained the temporally-aggregated interaction features $H$, we then feed them into a linear layer with a Sigmoid activation to generate parameters for $K$ proposal intervals. Each interval is represented by a tuple $(c_k, w_k)$, where $c_k, w_k$ are the normalized center and width, respectively. The corresponding temporal interval in the video timeline is therefore calculated as: $\hat{I}_k = [c_k - w_k/2, \ c_k + w_k/2]$.

We assign a ground-truth interval $\hat{I}_k^{gt}$ to each predicted interval $\hat{I}_k$ by maximizing the temporal intersection over union (IoU). If no overlap exists, we match based on the minimal temporal endpoint distance. The locator is trained with a localization loss, defined as follows:

$$\mathcal{L}_{\text{loc}} = -\frac{1}{K} \sum_{k=1}^{K} \log \left( \text{IoU}(\hat{I}_k, \hat{I}_k^{gt}) \right). \quad (2)$$

**Gaussian Mask.** To provide focused guidance for subsequent temporal focus classification (see § 3.1.3), we encode the predicted intervals as Gaussian masks $g = \{g_k^t\}_{k=1,t=1}^{K,T} \in \mathbb{R}^{K \times T}$, that assign soft frame-level weights within each interval. Here $g_k^t$ is the weight of the $t$-th frame for the $k$-th query. This approach explicitly captures the inherent temporal structure (start, peak, end) of relations (Zheng et al., 2022), enabling the classifier to emphasize in-

formative frames while suppressing irrelevant ones. Specifically, $g_k^t$ is computed as:

$$g_k^t = \frac{1}{\sqrt{2\pi} (w_k/\tau)} \exp\left( \frac{-(t/T - c_k)^2}{2(w_k/\tau)^2} \right), \quad (3)$$

where $\tau$ is a hyperparameter that controls the variance. To impose fine-grained structural constraints for the predicted intervals, we use corresponding ground-truth masks $g^{gt}$, constructed from ground-truth interval centers and widths, and minimize the mean squared error (MSE):

$$\mathcal{L}_{\text{mask}} = \frac{1}{K} \sum_{k=1}^{K} \text{MSE}\left(g_k, g_k^{gt}\right). \quad (4)$$

Finally, to encourage the queries to capture diverse temporal patterns, we apply a diversity loss (Lin et al., 2017):

$$\mathcal{L}_{\text{div}} = ||gg^\top - \lambda \mathbb{I}||_F^2, \quad (5)$$

where $||\cdot||_F$ denotes the Frobenius norm, and $\lambda$ is a hyperparameter that controls the degree of diversity.

### 3.1.3. TEMPORAL FOCUS CLASSIFIER

In this subsection, we introduce Temporal Focus Classifier, which utilizes a mask-guided attention mechanism, called *Temporal Focus Attention*, to predict relation categories. First, the entity pair feature $F \in \mathbb{R}^{T \times 2D}$ is projected to the attention query $Q_a \in \mathbb{R}^{T \times D}$, keys $K_a \in \mathbb{R}^{T \times D}$ and values $V_a \in \mathbb{R}^{T \times D}$. Then the attention map is calculated as $\mathbb{A} = \frac{Q_a K_a^T}{\sqrt{D}}$. To guide the attention toward the predicted intervals, we apply the corresponding Gaussian mask $g_k \in \mathbb{R}^T$ to the attention map via row-wise multiplication. The focusing attention feature for the $k$-th interval is given by:

$$\hat{F}_k = \text{Softmax}\left(\mathbb{A} \otimes g_k\right) \cdot V_a, \quad (6)$$

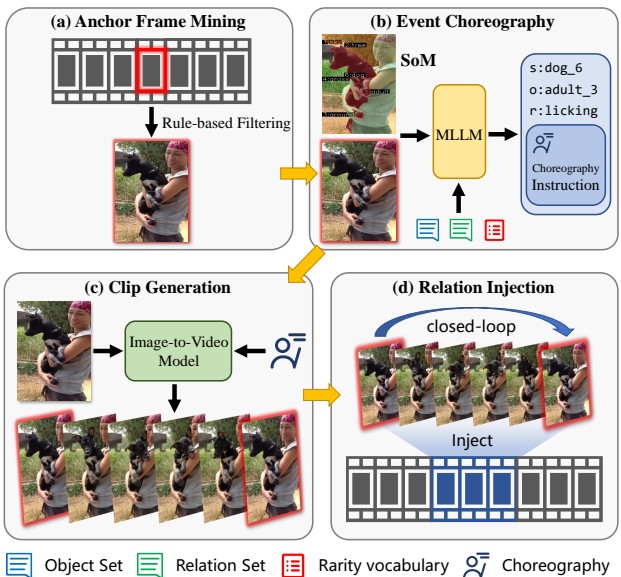

Figure 4. Overview of RGVA: (a) mine stable anchor frames via rule-based filtering; (b) use an MLLM (with SoM, object/relation sets, and a rarity-ranked vocabulary) to produce context-consistent choreography instructions for rare relations; (c) generate a short closed-loop interaction clip with an image-to-video model; and (d) inject the synthesized clip (with propagated annotations) back into the original training video.

where $\otimes$ denotes the row-wise multiplication. Feature $\hat{F}_k \in \mathbb{R}^{T \times D}$ is then transformed by an MLP and temporally max-pooled to generate relation scores $\hat{y}_k \in \mathbb{R}^C$, where $C = |\mathbb{C}^R|$ denotes the number of relation categories. For multi-label classification, we assign positive labels $y_k$ to ground-truth relations with an interval IoU$> 0.5$, and minimize the classification loss using binary cross-entropy (BCE):

$$\mathcal{L}_{\text{cls}} = \frac{1}{K} \sum_{k=1}^{K} \text{BCE}\left(\hat{y}_k, y_k\right), \tag{7}$$

### 3.1.4. MODEL TRAINING AND INFERENCE

**Training.** The overall objective combines localization loss $\mathcal{L}_{\text{loc}}$, mask alignment loss $\mathcal{L}_{\text{mask}}$, diversity loss $\mathcal{L}_{\text{div}}$, and classification loss $\mathcal{L}_{\text{cls}}$:

$$\mathcal{L}_{\text{total}} = \mathcal{L}_{\text{loc}} + \alpha_1 \mathcal{L}_{\text{mask}} + \alpha_2 \mathcal{L}_{\text{div}} + \alpha_3 \mathcal{L}_{\text{cls}}, \tag{8}$$

where $\alpha_1$, $\alpha_2$ and $\alpha_3$ are hyperparameters to balance losses.

**Inference.** We introduce a *Score-based Merging Strategy* to produce final predictions. Specifically, for a given relation category $c$, there are $K$ relation scores $y_k^c$. We select predicted intervals whose scores are within a margin $\delta$ of the maximum score: $\left\{\hat{I}_k \middle| \hat{y}_k^c \geq \hat{y}_{max}^c - \delta\right\}$. These selected intervals are merged to form the unified temporal span. Their average score is used as the confidence for retrieval.

## 3.2. Generative Video Augmentation

The second key component in SegPVSG is the RGVA. To address the long-tailed distribution of relations, this module implements an MLLM-guided Event Injection pipeline, which synthesizes video clips for rare relations and coherently inserts them into original video sequences. As shown in Figure 4, the pipeline consists of four stages. First, we identify anchor frames from the training set that provide a stable visual starting state for subsequent generation. Second, an MLLM generates context-consistent choreography instructions for rare relations. Third, guided by the instruction, an image-to-video model produces a short clip under a temporal closed-loop constraint (i.e., the first and last frames are identical to the anchor frame). Finally, the synthesized clips are annotated and inserted back into the existing training videos to provide additional supervision for tail categories. Details are provided in the following subsections.

### 3.2.1. ANCHOR FRAME MINING

To obtain appropriate generation seeds, we develop a four-step anchor frame mining process to extract anchor frames from the training set. First, we exclude frames located within annotated relation intervals to prevent temporal interference with existing relations. Second, we discard frames where target entities (i.e., animate entities such as adult, child, cat) are truncated by image boundaries to ensure spatial integrity. Third, we filter low-quality candidates by evaluating the sharpness of target regions. Specifically, a frame is retained only if the Laplacian variance of at least one target entity bounding box exceeds a predefined threshold. Finally, we apply interval-aware sampling to select a sparse yet diverse set of anchors for each video. More details can be found in the Appendix A.

### 3.2.2. RARE EVENT CHOREOGRAPHY

To synthesize rare relations that are compatible with the scene, we leverage an MLLM to reason over the anchor frame and generate rare event proposals. Given an anchor frame $\mathcal{F}$, MLLM generates rare event proposals by processing three additional inputs. First, the instance list (IDs and categories) and existing relations in $\mathcal{F}$ describe the current entities and interactions in text. Second, a Set-of-Mark (Yang et al., 2023c; Zhang et al., 2025) image $\mathcal{F}_{\text{SoM}}$ serves as a visual prompt that assigns a unique identifier to each object instance, facilitating visual grounding. Third, a rarity-ranked vocabulary $\mathcal{V}_{rank}$ organizes relations by their frequency to bias the model toward tail categories. By integrating these inputs with its commonsense knowledge, the MLLM formulates a rare event proposal $P = \langle (s, o, r), \mathcal{T} \rangle$. Here, the triplet $(s, o, r)$ identifies a potential relation $r$ between the subject $s$ and the object $o$. The choreography instruction $\mathcal{T}$ specifies how $s, o$ interact to realize relation $r$

from $\mathcal{F}$ and restore their original states. After each successful proposal, we update the relation frequencies in $\mathcal{V}_{rank}$ to encourage diverse events.

### 3.2.3. CLOSED-LOOP CLIP GENERATION

We employ an image-to-video model (Chen et al., 2025b) to synthesize the interaction clip $\mathcal{V} = \{\bar{\mathcal{F}}_t\}_{t=0}^{L_g}$ conditioned on the anchor frame $\mathcal{F}$ and instruction $\mathcal{T}$. Here, $L_g$ represents the total number of frames in the generated sequence. To minimize background drift and ensure temporal consistency, we enforce a temporal closed-loop constraint where the anchor frame $\mathcal{F}$ serves as both the first and last frames: $\bar{\mathcal{F}}_0 = \bar{\mathcal{F}}_{L_g} = \mathcal{F}$. By imposing this constraint, the generator focuses on the relative motion between the subject $s$ and object $o$ while maintaining a static background. This results in an atomic interaction unit that can be coherently integrated into the original video stream.

### 3.2.4. ANNOTATION PROPAGATION AND INTEGRATION

We generate panoptic annotations for the synthesized clips to integrate them into the PVSG training pipeline. Specifically, we use the ground-truth mask of the anchor frame $\mathcal{F}$ for both the first and the last frames. For the intermediate frames, we employ SAM2 (Ravi et al., 2025) to propagate the masks to ensure temporal coherence. We then insert these frames into the original video stream at the anchor timestamp as an additional relation segment, and update the associated annotations to maintain temporal consistency. This method effectively augments the dataset while preserving the original instance identities and scene context.

## 4. Experiments

### 4.1. Experiment Settings

**Datasets.** We evaluate our method on OpenPVSG (Yang et al., 2023b), which contains 400 video clips (338 for training and 62 for testing) with an average duration of 77 seconds. OpenPVSG provides 7.6K object mask tracks (126 categories), along with 4548 relation triples (57 relation types). We additionally evaluate on ImageNet-VidVRD (Shang et al., 2017), a VidSGG benchmark with 1000 videos (800 for training and 200 for testing). It contains 35 object categories and 132 relation types.

**Data Augmentation Settings.** For rare relation video clip generation (see pipeline in § 3.2), anchor frames and their panoptic annotations are extracted from the OpenPVSG dataset (Yang et al., 2023b). We use Seed1.6 for rare-event proposal generation and Seedance 1.5 Pro (Chen et al., 2025b) for video synthesis. Each generated clip is 5 seconds long and rendered at the nearest supported resolution to its corresponding anchor frame. We only augment the training videos in place, while leaving the test splits unchanged.

**Evaluation Metrics.** Following prior works (Yang et al., 2023b; Nguyen et al., 2025b), we report recall at $K$ (R@K) and mean recall at $K$ (mR@K) over the top-$K$ predicted relation triples. R@K measures overall retrieval performance across all relations, whereas mR@K averages recall over relation categories and therefore better reflects performance on rare relations. For a predicted triplet to be counted as correctly recalled, both of the following conditions should be satisfied: (1) the correct category labels of the subject, object, and relation; (2) the volume-IoU (vIoU) between the predicted mask tubelet and ground-truth mask tubelet reaching a specified threshold of 0.5 or 0.1.

**Implementation Details.** Following prior works (Yang et al., 2023b; Nguyen et al., 2025b), we use identical segmentation modules across all experiments to ensure fair comparisons, including both IPS+T and VPS settings. For IPS+T, we adopt a fine-tuned Mask2Former (Cheng et al., 2022) for frame-level panoptic segmentation and a pretrained UniTrack (Wang et al., 2021) to construct object tubes. For VPS, we use Video K-Net (Li et al., 2022a) integrated into the Mask2Former framework for video panoptic segmentation. Both segmentation models are built on Mask2Former with a ResNet-50 (He et al., 2016) backbone. They are trained for 8 epochs and kept frozen during TFN training. We optimize TFN with AdamW (Loshchilov & Hutter, 2017) using a learning rate of $1e^{-4}$ and weight decay of $1e^{-2}$. The model is trained for 100 epochs on an RTX 4090. For RGVA-augmented training (full SegPVSG), we follow the same IPS+T setting to obtain consistent panoptic masks and object tubes. The feature dimension is fixed to $D = 256$. Unless otherwise specified, we set $K = 8$, $\tau = 9$, $\lambda = 0.15$, $\alpha_1 = 0.5$, $\alpha_2 = 2.5$, and $\alpha_3 = 0.5$. On ImageNet-VidVRD (Shang et al., 2017), we use object tubes extracted by MEGA (Chen et al., 2020), set $D = 512$, and use a learning rate of $5e^{-5}$, while keeping all other settings consistent with those on OpenPVSG.

**Compared Methods.** Under IPS+T and VPS settings, we compare our method against representative PVSG baselines, including two-stage PVSG variants (Yang et al., 2023b) (Vanilla, Handcrafted Filter, Transformer, and 1D Conv) and the motion-aware extension (Nguyen et al., 2025b), which enhances Transformer/1D Conv backbones with motion-aware contrastive learning. We also include UNO (Le et al., 2026), a one-stage end-to-end model that jointly performs panoptic segmentation and relation prediction. For UNO, we report the ResNet-50 variant to match the backbone of our Mask2Former-based segmentation models.

### 4.2. Main Result

**Evaluation on OpenPVSG.** Table 1 reports the comparative results on OpenPVSG. Overall, the proposed SegPVSG achieves state-of-the-art performance across all settings,

*Table 1.* Comparison between SegPVSG and other methods on the OpenPVSG dataset in both IPS+T and VPS settings.

| Method | | vIoU threshold = 0.5 | | | vIoU threshold = 0.1 | | |
|---|---|---|---|---|---|---|---|
| | | R/mR@20 | R/mR@50 | R/mR@100 | R/mR@20 | R/mR@50 | R/mR@100 |
| One-stage | UNO ResNet-50 | 6.23 / 5.60 | 7.37 / 6.84 | 8.65 / 8.21 | 13.83 / 11.65 | 19.27 / 15.94 | 24.63 / 19.99 |
| IPS+T | Vanilla | 3.04 / 1.35 | 4.61 / 2.94 | 5.56 / 3.33 | 8.28 / 5.68 | 14.47 / 9.92 | 18.24 / 11.84 |
| | Handcrafted filter | 2.52 / 1.72 | 3.77 / 2.36 | 4.72 / 2.79 | 8.07 / 5.61 | 13.42 / 8.27 | 16.46 / 10.11 |
| | Transformer | 3.88 / 2.81 | 5.66 / 4.12 | 6.18 / 4.44 | 9.01 / 6.69 | 14.88 / 11.28 | 17.51 / 13.20 |
| | Convolution | 3.88 / 2.55 | 5.24 / 3.29 | 6.71 / 5.36 | 10.06 / 8.98 | 14.99 / 12.21 | 18.13 / 15.47 |
| | Motion-aware Transformer | 3.98 / 2.98 | 5.97 / 4.20 | 7.44 / 5.15 | 10.59 / 9.56 | 16.98 / 12.3 | 22.33 / 17.47 |
| | Motion-aware Convolution | 4.51 / 3.56 | 6.08 / 4.38 | 7.76 / 5.86 | 11.43 / 9.57 | 17.30 / 13.13 | 22.85 / 17.48 |
| | **SegPVSG (Ours)** | **5.45 / 9.13** | **9.54 / 12.74** | **13.63 / 15.91** | 11.95 / **16.54** | 20.13 / **22.98** | **28.72 / 28.92** |
| | **SegPVSG (w/o RGVA)** | 5.14 / 6.67 | 9.12 / 10.15 | 12.58 / 12.87 | **12.16** / 13.20 | **21.38** / 21.58 | 27.57 / 26.45 |
| VPS | Vanilla | 0.21 / 0.10 | 0.21 / 0.10 | 0.31 / 0.18 | 6.29 / 3.04 | 9.64 / 6.74 | 12.89 / 9.60 |
| | Handcrafted filter | 0.42 / 0.13 | 0.52 / 0.50 | 0.94 / 0.92 | 5.24 / 2.84 | 7.65 / 7.14 | 9.64 / 8.22 |
| | Transformer | 0.42 / 0.61 | 0.73 / 0.76 | 1.05 / 0.92 | 6.50 / 5.75 | 9.64 / 8.25 | 12.26 / 9.51 |
| | Convolution | 0.42 / 0.25 | 0.63 / 0.67 | 0.63 / 0.67 | 8.07 / 7.84 | 11.01 / 9.78 | 12.89 / 10.77 |
| | Motion-aware Transformer | 0.63 / 0.83 | 1.05 / 0.76 | 1.05 / 0.76 | 6.71 / 6.94 | 10.27 / 8.68 | 13.42 / 12.09 |
| | Motion-aware Convolution | 0.84 / 0.98 | 1.26 / 1.22 | 1.26 / 1.22 | 8.18 / 8.00 | 12.90 / 11.47 | 14.22 / 13.59 |
| | **SegPVSG (Ours)** | **1.78 / 2.26** | **2.52 / 2.67** | **3.04 / 4.82** | **9.22 / 9.01** | 15.30 / **15.45** | 19.29 / 20.93 |
| | **SegPVSG (w/o RGVA)** | **1.78** / 1.52 | **2.52** / 2.16 | **3.04** / 2.79 | 8.39 / 8.45 | **15.72** / 15.33 | **22.01 / 21.63** |

demonstrating superior robustness and balanced retrieval capabilities. Under the IPS+T setting, SegPVSG significantly outperforms all prior two-stage baselines, improving R/mR@100 by +5.87/+10.05 over the strongest competitor at vIoU=0.5. Compared to the one-stage method UNO, SegPVSG surpasses it on R@50, R@100, and all mean Recall (mR) metrics. This indicates that while UNO favors a concentrated set of high-confidence predictions, our method retrieves a more diverse and comprehensive set of relations. In the challenging VPS setting, SegPVSG maintains high performance, proving its resilience to mask noise. Additionally, the base model (w/o RGVA) alone surpasses all prior two-stage methods, confirming that explicitly localizing relation intervals is crucial for robust recognition, especially under noisy conditions (VPS).

**Long-tail Distribution Analysis.** As shown in Table 1 and Figure 5, RGVA substantially alleviates data imbalance. Specifically, it boosts mR@100 by +3.04, proving that augmenting rare-relation segments effectively enhances the recognition of tail predicates. We further report the relation-wise Normalized Recall@20 (Recall@20 divided by maximum attainable recall) under IPS+T. TFN consistently outperforms the Transformer baseline on the majority of relation categories. Moreover, the full SegPVSG further improves the performance, particularly in rare relations.

**Evaluation on ImageNet-VidVRD.** To assess generalization beyond pixel-level supervision, we compare with state-of-the-art methods on ImageNet-VidVRD, using a bounding box vIoU threshold of 0.5. TFN demonstrates strong adaptability, achieving superior performance despite the coarser annotations. As shown in Table 2, our method outperforms these specialized competitors, confirming that the temporal

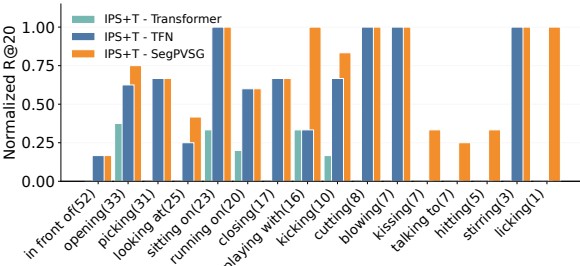

*Figure 5.* Normalized R@20 across relation categories for different methods. Numbers in parentheses are sample counts.

*Table 2.* Performance on the ImageNet-VidVRD dataset.

| Method | R/mR@20 | R/mR@50 | R/mR@100 |
|---|---|---|---|
| Transformer | 8.19 / 5.18 | 11.71 / 8.19 | 14.50 / 11.14 |
| Convolution | 11.79 / 7.80 | 17.61 / 11.70 | 22.24 / 15.02 |
| BIG-C | 13.00 / 9.46 | 18.13 / 14.32 | 24.27 / 18.98 |
| VrdONE | **13.96** / 8.51 | 18.62 / 11.52 | 21.58 / 13.03 |
| **TFN (Ours)** | 13.09 / **10.01** | **21.71 / 17.64** | **29.18 / 22.66** |

localization mechanism of TFN remains robust even when constrained to bounding box tubelets.

### 4.3. Ablation Study

We perform ablation studies on OpenPVSG under the IPS+T setting. More ablations and visualization results are provided in the Appendices B and C.

**Effect of Gaussian-masked Focusing.** To examine the role of temporal focusing in relation recognition, we replace the Temporal Focus Attention module with a 1D convolutional layer, a Transformer encoder, and a binary hard mask. As

Table 3. Ablation on temporal modeling strategies.

| Backbone | R/mR@20 | R/mR@50 | R/mR@100 |
|----------|---------|---------|----------|
| Convolution | 4.61 / 5.05 | 7.02 / 7.42 | 9.96 / 11.57 |
| Transformer | 3.35 / 4.14 | 7.23 / 9.28 | 11.53 / 12.23 |
| Hard mask | 4.72 / 5.24 | 8.60 / 9.74 | 11.95 / 12.11 |
| **TFN (Ours)** | **5.14 / 6.67** | **9.12 / 10.15** | **12.58 / 12.87** |

Table 4. Ablation study on relation temporal localization capability.

| Method | R@Inf | w R@Inf | soft R@Inf |
|--------|-------|---------|------------|
| Vanilla | 6.60 | 22.75 | 4.81 |
| Handcrafted filter | 7.34 | 25.15 | 5.16 |
| Transformer | 9.01 | 27.57 | 6.57 |
| Convolution | 10.38 | 27.78 | 7.43 |
| **TFN (Ours)** | **21.49** | **45.91** | **16.67** |

Table 5. Different methods with RGVA.

| Method | R/mR@20 | R/mR@50 | R/mR@100 |
|--------|---------|---------|----------|
| Transformer | 3.88 / 2.81 | 5.66 / 4.12 | 6.18 / 4.44 |
| Transformer + RGVA | 3.88 / 3.20 | 5.45 / 4.55 | 7.13 / 7.19 |
| Convolution | 3.88 / 2.55 | 5.24 / 3.29 | 6.71 / 5.36 |
| Convolution + RGVA | 3.88 / 3.75 | 6.08 / 5.71 | 7.02 / 6.01 |
| SegPVSG (w/o RGVA) | 5.14 / 6.67 | 9.12 / 10.15 | 12.58 / 12.87 |
| **SegPVSG (Ours)** | **5.45 / 9.13** | **9.54 / 12.74** | **13.63 / 15.91** |

Table 6. Comparison with long-tail learning baselines on Open-PVSG under the IPS+T setting.

| Setting | RGVA | Reweight | R/mR@50 | R/mR@100 |
|---------|------|----------|---------|----------|
| TFN | | | 9.43 / 5.12 | 12.79 / 8.79 |
| + RGVA | ✓ | | 9.33 / 5.15 | 13.00 / 11.51 |
| + Reweight | | ✓ | 9.12 / 10.15 | 12.58 / 12.87 |
| + Both | ✓ | ✓ | **9.54 / 12.74** | **13.63 / 15.91** |

shown in Table 3, TFN achieves the best performance across all metrics. This improvement is mainly attributed to the Gaussian mask, which imposes soft temporal constraints that highlight the predicted interval while remaining tolerant to boundary noise. In contrast, the hard mask is sensitive to imperfect boundaries, and generic temporal encoders lack an explicit mechanism to suppress irrelevant frames, leading to less robust relation recognition.

**Comparison of Relation Temporal Localization Capability.** We benchmark the Relation Interval Locator against PVSG baselines using Recall without top-K restrictions (R@Inf) and its soft variant, aiming to isolate temporal localization quality from classification rankings. As shown in Table 4, TFN consistently outperforms the baselines, demonstrating the superior precision of our query-based interval prediction. This advantage is particularly evident in soft R@Inf, which measures fine-grained overlap by aggregating continuous vIoU scores, confirming that TFN aligns relation boundaries more accurately than previous approaches.

**Collaboration of TFN and RGVA.** We evaluate the synergy between TFN and RGVA by training various methods using RGVA-augmented videos. As shown in Table 5, TFN benefits significantly more than the baselines, demonstrating superior compatibility with the generated data. This suggests that while RGVA enriches supervision for rare relations, TFN's interval-aware focusing better aligns learning to the intended interaction segment, thereby converting additional, potentially noisy augmented data into more balanced improvements across relation categories.

**Comparison with Long-tail Learning Baselines.** To verify that RGVA is not merely equivalent to simple class rebalancing, we compare it with loss reweighting in Table 6. Note that our full SegPVSG adopts loss reweighting by default; in this ablation, "TFN" denotes the variant without both RGVA and loss reweighting. As shown in Table 6, loss reweighting

improves mean recall but slightly reduces overall recall, indicating a trade-off between balanced recognition and overall retrieval. In contrast, RGVA improves mean recall by introducing additional visual-temporal evidence for rare relations while maintaining competitive overall recall. Combining RGVA with loss reweighting achieves the best results, showing that the two strategies are complementary: reweighting adjusts the optimization objective, whereas RGVA directly alleviates data scarcity.

**Fine-grained Long-tail Analysis.** We further split relation categories into Head, Body, and Tail groups according to their training frequencies and report R/mR@100 in Table 7. TFN improves over the Transformer baseline across all splits, showing that explicit interval localization benefits relations with different frequencies. SegPVSG further boosts the Tail split while maintaining strong performance on Head and Body relations. This indicates that RGVA improves rare-relation recognition without simply sacrificing frequent-category performance.

**Objective Quality Assessment of RGVA-generated Clips.** We evaluate the generated clips using Qwen3-VL-Embedding (Li et al., 2026) for video-text similarity and DOVER (Wu et al., 2023) for no-reference video quality assessment. As shown in Table 8, synthetic clips achieve scores comparable to real training videos, suggesting reasonable semantic alignment and perceptual quality. We also filter generated clips with mismatched subject-object pairs, severe temporal artifacts, or incorrect interactions. These results indicate that RGVA provides useful augmentation data rather than introducing uncontrolled noise.

### 4.4. Qualitative Analysis

**Predictions Visualization.** Figure 6 provides qualitative comparisons. In Figure 6(a), the relation between *adult_4* and *ground_2* changes over time from "standing on" to

*Table 7.* Fine-grained long-tail analysis on Head, Body, and Tail relation splits. We report R/mR@100 under the IPS+T setting.

| Method | Head (30%) | Body (40%) | Tail (30%) |
|---|---|---|---|
| Transformer | 6.56 / 6.72 | 3.03 / 2.46 | 4.76 / 4.90 |
| TFN | 12.85 / 15.72 | 14.55 / 15.57 | 7.14 / 6.86 |
| SegPVSG | **13.12 / 15.97** | **16.57 / 18.43** | **9.52 / 12.75** |

*Table 8.* Objective quality assessment of real and RGVA-generated videos. Global and Inter-class denote Qwen3-VL-based video-text similarity scores, while Aesthetic, Technical, and Overall denote DOVER quality scores. We report mean±std for all metrics.

| Video Set | Global | Inter-class |
|---|---|---|
| Real Videos | 0.582±0.093 | 0.572±0.074 |
| Synthetic Videos | 0.549±0.088 | 0.555±0.054 |

| Video Set | Aesthetic | Technical / Overall |
|---|---|---|
| Real Videos | 93.5±13.0 | 5.6±1.7 / 43.9±16.9 |
| Synthetic Videos | 93.0±10.7 | 7.5±1.8 / 44.9±14.6 |

"walking on". While the prior method only detects the static "standing on" relation, SegPVSG accurately captures the transition with precise temporal boundaries, demonstrating its capability in handling dynamic relations in PVSG. In Figure 6(b), our approach successfully localizes all interaction intervals, better handling complex temporal dynamics and recurrent relations. Notably, SegPVSG performs robustly across both third-person and egocentric perspectives.

**Generation Visualization.** Figure 7 illustrates the synthesis capability of RGVA. Specifically, the image-to-video model generates interaction segments for rare relations (e.g., drinking from, catching), guided by choreography instructions that combine fine-grained visual descriptions with multi-stage actions. Crucially, the temporal closed-loop constraint ensures identical start and end frames (marked by orange borders), maintaining background stability alongside coherent temporal dynamics. These clips are seamlessly injected into the original videos and serve as the new training set.

## 5. Conclusion

This paper presents SegPVSG, a temporal-segment-aware framework for the PVSG task, which is built upon two key components, TFN and RGVA. The TFN explicitly localizes interaction intervals and reduces irrelevant contextual interference, thus enhancing relation modeling in PVSG. In addition, the RGVA synthesizes diverse training clips for rare relations, alleviating the long-tailed distribution issue and consequently improving the model's generalization. Extensive experiments on OpenPVSG and ImageNet-VidVRD demonstrate that our method achieves significant improvements in relation recall, particularly for tail categories.

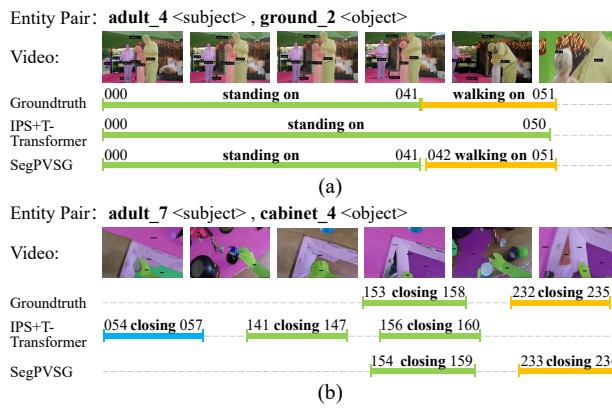

(a)

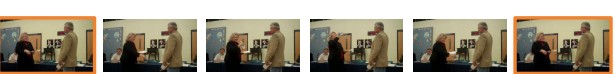

(b)

*Figure 6.* Qualitative comparison between our method and IPS+T-Transformer (Yang et al., 2023b).

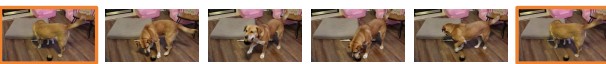

**Target Triplet**: (**adult_7** <subject>, **water_3** <object>, **drinking from**)
**Choreography Instruction**: A blonde woman in a black blazer and red top stands behind a table, holding a white paper in her left hand. First, she extends her right hand, picks up a water bottle from the table, and brings it to her lips to drink. Then, she places the bottle back on the table. Finally, she returns her right hand to her side, resuming her initial static pose.

**Target Triplet**: (**dog_3** <subject>, **toy_6** <object>, **catching**)
**Choreography Instruction**: A large brown dog starts in a static position. First, it lowers its head and bats the small black and brown toy upward with its paw. Then, it catches the toy in its mouth before releasing it, letting the toy fall back to the floor. Finally, the dog retracts its paw and raises its head, returning to its initial pose.

*Figure 7.* Qualitative examples of generation results by RGVA. Orange borders mark the anchor frames, indicating identical start and end states.

## Acknowledgements

This work was supported by National Natural Science Foundation of China (No.62576139, 62176093), National Key Research and Development Program of China (No.2023YFC3502900).

## Impact Statement

Our method uses generative models to synthesize context-aware video augmentation data, which constitutes a dual-use technology. Although it is designed to improve video understanding, it could be misused to produce misinformation or deceptive video content. Moreover, because the method relies on large-scale pretrained generative models, the synthesized data may inherit potential biases or stereotypes from these models and their source data, which could propagate to downstream systems. Finally, even when datasets are obtained and used in compliance with their licenses, training with generated data remains an actively debated societal issue and should be approached with caution.

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

# Appendix Overview

In this supplementary material, we present additional details and results that could not be included in the main paper due to space limitations. It includes implementation details of RGVA, supplementary ablation studies, and additional qualitative visualizations.

- Appendix A provides more details of RGVA.

- Appendix B reports supplementary experimental results.

- Appendix C presents additional qualitative visualizations and failure case analysis.

# A. Additional Details about RGVA

## A.1. More Implementation Details

**About Anchor Frame Mining**. In the OpenPVSG annotations, each relation is represented as a temporal interval. To avoid breaking the original relations when synthesizing clips, we exclude frames that fall inside any annotated relation interval when selecting anchor frames. However, we ignore relations whose duration exceeds 80% of the corresponding video length, since such long-term relations are typically stable and unlikely to be affected by localized edits; otherwise, this filtering step would leave too few candidate frames. After applying all filters, the remaining candidate frames usually form one or several contiguous segments. We therefore adopt an interval-aware sampling strategy: we uniformly sample a segment-dependent number of frames from each contiguous segment, which maintains a sufficient number of anchors while encouraging diversity among the mined anchor frames.

**About Generation Pipeline Prompt.** To generate valid relation triplets and actionable "round-trip" choreography, our prompt conditions the model on two visual inputs (a raw frame for physical evidence and an ID mask for object correspondence), together with the available object list, existing ground-truth relations, and a rarity-prioritized relation vocabulary. The MLLM proposes candidate triples and filters them according to (1) logical compatibility with existing relations and (2) visual plausibility in the raw frame (e.g., proximity, occlusion, free hands/posture), and selects the highest-priority feasible relation. It then produces an ID-free, kinematic start–action–return video description and outputs the justification, the new triplet, and the choreography instruction. The full prompt is in Figure 13 at the end of the appendix (Lin et al., 2026).

## A.2. RGVA Generation Statistics and Filtering

We target rare relations and synthesize 566 candidate video clips in total. After filtering, we retain 386 clips (68%), with the relation-wise augmentation summarized in Fig-

ure 8. Some rare relations receive only a small number of generated clips because suitable scenes are inherently scarce under our constraints. For example, generating a plausible "watering" clip typically requires a person, a kettle (or watering can), and plants to co-occur in the scene, while the relevant entities also remain available for interaction, which rarely happens in practice.

We filter generated clips using the following criteria: (1) the interacting subject/object in the generated clip does not match the pre-specified target pair; (2) objects abruptly appear or disappear over time, indicating inconsistent content; (3) the performed interaction deviates substantially from the target relation (i.e., the choreography does not realize the intended relation).

# B. Additional Ablation Studies

We present the ablation experiment results on the Open-PVSG datasets with the setting of IPS+T as the segmentation module. We report Recall at K (R@K) and mean Recall at K (mR@K) metrics under the vIoU threshold of 0.5. The experimental settings remained consistent with those presented in the main paper.

## B.1. Effect of the Number of Temporal Queries

To examine the impact of the number of learnable temporal queries, we vary $K$ over values $\{2, 4, 6, 8, 10, 12\}$ and compare their performance. As shown in Figure 9, a moderate number of temporal queries yields the best results. $K = 8$ gives the highest recall and $K = 10$ provides the best mean recall. When $K$ is too small, the queries cannot cover all potential relation intervals, leading to missed detections. Conversely, a large $K$ exceeds the actual demand of the task, causing conflicts between localization and diversity, which leads to unstable training. Considering the trade-off between accuracy and efficiency, we choose $K = 8$ as a practical setting.

## B.2. Impact of Losses for Relation Interval Locator

We conducted an ablation study to examine the importance of each loss item in the Relation Interval Locator. The results in Table 9 reveal that $\mathcal{L}_{\mathrm{loc}}$ is the most crucial loss item under this setting, as it directly constrains the accuracy of predicted relation intervals, ensuring reliable temporal grounding for downstream relation classification. Moreover, both $\mathcal{L}_{\mathrm{mask}}$ and $\mathcal{L}_{\mathrm{div}}$ contribute effectively to enhancing model performance. The former provides fine-grained structural guidance over the temporal intervals, while the latter encourages the predicted interval masks to capture diverse temporal patterns. Combining these three losses can yield better performance.

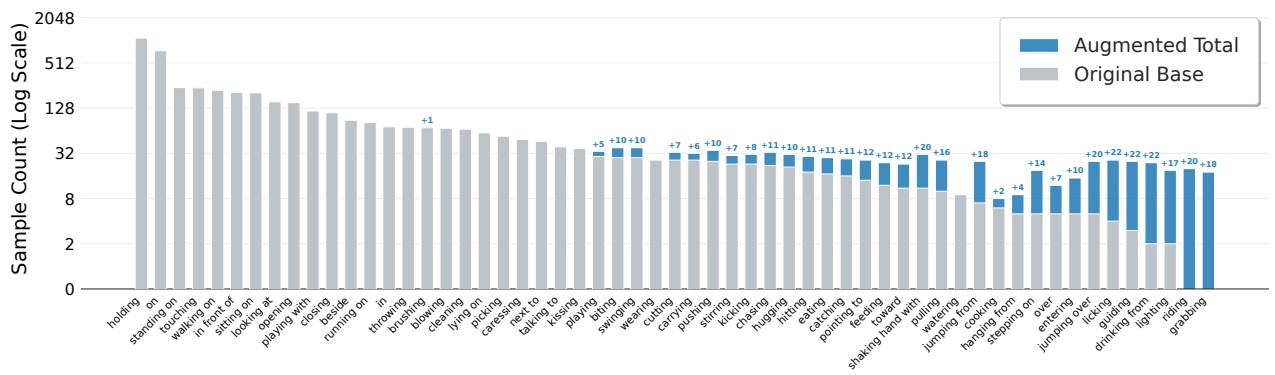

*Figure 8.* The enhancement effect of RGVA for the OpenPVSG trainset.

*Table 9.* Ablations of each loss item for TFN.

| $\mathcal{L}_{loc}$ | $\mathcal{L}_{mask}$ | $\mathcal{L}_{div}$ | R/mR@20 | R/mR@50 | R/mR@100 |
|---|---|---|---|---|---|
| | ✓ | ✓ | 1.36 / 1.53 | 3.25 / 2.52 | 3.98 / 3.51 |
| ✓ | | ✓ | 4.72 / 5.76 | 8.70 / 8.11 | 12.05 / 10.62 |
| ✓ | ✓ | | 4.61 / 5.20 | 8.91 / 8.60 | 12.05 / 10.44 |
| ✓ | ✓ | ✓ | **5.14 / 6.67** | **9.12 / 10.15** | **12.58 / 12.87** |

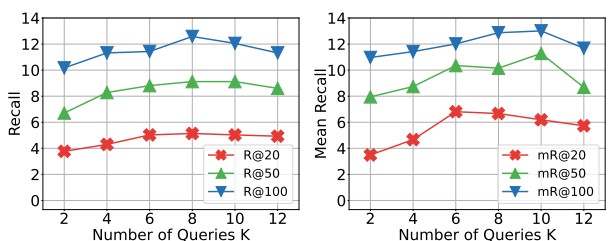

*Figure 9.* Ablation results on different numbers of queries.

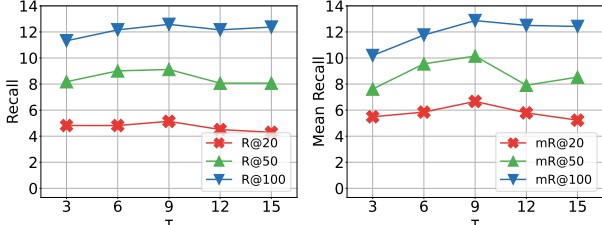

*Figure 10.* Ablation of the Gaussian mask variance parameter $\tau$ for TFN.

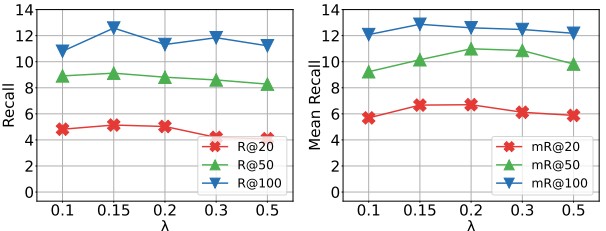

*Figure 11.* Ablation of the diversity weight $\lambda$ for TFN.

### B.3. Effect of the Gaussian Mask Sharpness

We investigate how the shape of the Gaussian mask affects relation modeling. The shape is controlled by the variance parameter $\tau$ in Equation (3), where a larger $\tau$ results in a sharper shape. We vary $\tau$ within the set $\{3, 6, 9, 12, 15\}$ and report the corresponding results in Figure 10. As shown, the model achieves the best performance when $\tau = 9$, suggesting a balance between precise temporal localization and tolerance to slight misalignment. Both overly sharp and overly flat masks degrade performance, highlighting the importance of appropriately modeling the temporal structure of relation intervals.

### B.4. Effect of Diversity Degree for Gaussian Masks

We conduct an ablation study to investigate the effect of the diversity regularization weight $\lambda$ in Equation (5), which controls the strength of the diversity regularization applied to the predicted Gaussian masks. We vary $\lambda$ in the range $\{0.1, 0.15, 0.2, 0.3, 0.5\}$ and report the results in Figure 11. As shown, performance improves as $\lambda$ increases from 0.1 to 0.15 or 0.2, then declines when $\lambda$ becomes larger. These

results suggest that an appropriate $\lambda$ promotes diversity among temporal queries, thereby enhancing the ability of the model to cover varied relation patterns.

### B.5. Effect of Entity Feature Representations

To assess the effect of different entity feature representations, we perform three ablation settings: (1) directly using the query in Mask2Former as the entity feature (Yang et al., 2023b), (2) using only the visual features obtained via mask pooling, and (3) our full method, which augments the visual features with category semantic embeddings. As reported in Table 10, visual features lead to clear improvements over query features, as they provide essential cues for interaction interval localization. Furthermore, combining visual features with category semantic embedding brings additional improvements, since entity categories offer strong priors for relation prediction.

*Table 10.* Ablation study comparing different entity features.

| Entity Feature | R/mR@20 | R/mR@50 | R/mR@100 |
|---|---|---|---|
| Query Feature | 4.51 / 5.51 | 7.86 / 7.33 | 12.16 / 10.74 |
| Visual Feature | 4.93 / 5.94 | 8.91 / 9.13 | 11.84 / 10.98 |
| **Vis + Semantic** | **5.14 / 6.67** | **9.12 / 10.15** | **12.58 / 12.87** |

*Table 11.* Ablation study on score aggregation strategies used in the Score-based Merging Strategy.

| Score Strategy | R/mR@20 | R/mR@50 | R/mR@100 |
|---|---|---|---|
| Max Score | 4.82 / 4.53 | 7.65 / 6.22 | 9.12 / 8.11 |
| Ave Score | **5.14 / 6.67** | **9.12 / 10.15** | **12.58 / 12.87** |

*Table 12.* Ablation study on the supervision source for the Temporal Focus Classifier

| supervision | R/mR@20 | R/mR@50 | R/mR@100 |
|---|---|---|---|
| Predicted Intervals | 4.82 / 4.53 | 7.65 / 6.22 | 9.12 / 8.11 |
| Ground-Truth | **5.14 / 6.67** | **9.12 / 10.15** | **12.58 / 12.87** |

*Table 13.* Oracle perception analysis using ground-truth entity labels on OpenPVSG under the IPS+T setting.

| Setting | R/mR@20 | R/mR@50 | R/mR@100 |
|---|---|---|---|
| TFN | 5.14 / 6.67 | 9.12 / 10.15 | 12.58 / 12.87 |
| TFN + GT | 13.10 / 15.10 | 23.38 / 24.32 | 32.39 / 31.75 |
| SegPVSG | 5.45 / 9.13 | 9.54 / 12.74 | 13.63 / 15.91 |
| SegPVSG + GT | **13.63 / 15.62** | **26.31 / 27.80** | **41.09 / 39.36** |

## B.6. Comparison of Score Aggregation Strategies

We conduct an ablation study to compare two score aggregation strategies used to compute the relation score for retrieval in our Score-based Merging Strategy. Specifically, we evaluate a variant of our method where the average score is replaced by the maximum score. As shown in Table 11, the average score achieves better performance on most metrics. These results demonstrate that averaging provides a more robust and balanced estimation of relation confidence, as it effectively integrates evidence from multiple intervals. In contrast, the maximum score tends to overemphasize single predictions, making the relation score more sensitive to noise.

## B.7. Analysis of Training Supervision Strategies

We conducted an ablation study training the Temporal Focus Classifier using predicted intervals instead of ground-truth annotations. As shown in Table 12, while this strategy theoretically mitigates the train-test distribution shift, it yields inferior overall performance, particularly in Mean Recall. We attribute this degradation to supervision starvation induced by the long-tail nature of PVSG relations. Since tail categories are inherently challenging to localize, relying solely on predicted intervals results in frequent false negatives and effectively starves the classifier of essential training signals for rare classes. Consequently, our ground-truth-based approach proves superior by decoupling representation learning from localization errors, ensuring robust supervision across the entire distribution. Future work may explore hybrid curriculum strategies to bridge the domain gap without compromising tail performance.

## B.8. Oracle Perception Analysis

We further analyze the influence of upstream perception quality by replacing predicted entity labels with ground-truth labels during evaluation. As shown in Table 13, both TFN and SegPVSG benefit significantly from oracle perception, indicating that entity recognition errors remain an important bottleneck for PVSG. Importantly, SegPVSG

still consistently outperforms TFN under the oracle setting, demonstrating that the improvement brought by RGVA is complementary to better perception and is not merely caused by differences in upstream entity predictions.

## B.9. Computational Overhead of RGVA

RGVA is an offline augmentation pipeline and therefore introduces no additional inference-time overhead. We report the offline generation cost in Table 14 and the training overhead in Table 15. The total generation cost is moderate because event proposal and video synthesis are performed only once before training. After augmentation, the average video length and the number of relation instances increase, leading to a small increase in training time per epoch. Nevertheless, the inference procedure remains unchanged, since RGVA only modifies the training data.

## B.10. Computational Complexity

We compare the computational complexity of different temporal relation modules in Table 16. TFN introduces a moderate increase in parameters and FLOPs compared with Transformer and Convolution baselines, due to the temporal query decoder and Gaussian-mask-guided attention. However, it maintains comparable inference speed while achieving substantially better R/mR@100. These results show that TFN provides a favorable accuracy-efficiency trade-off.

## C. More Visualization Results

### C.1. Qualitative Examples of SegPVSG

**Visualization Results.** Figure 12 shows some qualitative examples of SegPVSG. As shown in Figure 12(a), SegPVSG retrieves multiple child-ball relations, including playing with, kicking, and chasing, whereas the Transformer baseline misses kicking and chasing, demonstrating its capability in handling dynamic relations in PVSG. In Figure 12(b), our approach successfully detects all interaction intervals with accurate temporal boundaries, highlighting that Seg-

*Table 14.* Offline generation cost of RGVA. The cost is incurred only once before model training.

| Stage | Model | API Calls | Avg. Cost/Call($) | Total Cost($) |
|---|---|---|---|---|
| Instruction Inference | Seed-1.6 | 586 | 0.004 | 2.366 |
| Video Generation | Seedance-1.5-Pro | 570 | 0.058 | 32.852 |

*Table 15.* Training overhead introduced by RGVA on OpenPVSG under the IPS+T setting.

| Training Set | Videos | Relations | Avg. Length | Time/Epoch |
|---|---|---|---|---|
| Original Set | 338 | 4729 | 75.08s | 250.43s |
| Augmented Set | 338 | 5115 | 80.56s | 272.43s |

*Table 16.* Computational complexity and efficiency comparison of different temporal relation modules.

| Method | R/mR@100 | Params | FPS | GFLOPs/video |
|---|---|---|---|---|
| Transformer | 6.18 / 4.44 | 26.55M | **897.74** | **94.95** |
| Convolution | 6.71 / 5.36 | **15.76M** | 802.52 | 151.86 |
| TFN | **12.58 / 12.87** | 32.31M | 834.12 | 157.48 |

PVSG can better handle complex temporal dynamics and recurrent relations. In Figure 12(c), SegPVSG again shows robustness by successfully retrieving the fine-grained relation *blowing*. This example demonstrates that our method effectively captures subtle and transient interactions. Notably, our approach performs well in both third-person and egocentric views, reflecting its robustness across diverse visual perspectives.

**Analysis of Failure Cases.** Despite the overall effectiveness of SegPVSG, some failure cases reveal challenges in the upstream perception modules. As shown in Figure 12(a), the tracker fails to maintain consistent instance IDs for the same subject (child) and object (ball) across frames. Consequently, SegPVSG splits the relation *playing with* into two separate segments. In Figure 12(b), segmentation errors lead to missing masks for the object (floor) in the middle of the relation, resulting in a fragmented prediction for the *standing on* relation. These cases highlight the dependency of relation reasoning on robust entity tracking and segmentation, suggesting potential benefits from tighter integration with these components.

## D. Discussion

**Failure Cases of RGVA.** Although the closed-loop design and filtering strategy improve the reliability of RGVA, some generated clips still fail to satisfy the desired relation constraints. We observe three typical failure modes: (1) subject-object mismatch, where the generated interaction involves a different instance from the pre-specified target pair; (2) temporal inconsistency, where objects abruptly appear, disappear, or change identity across frames; and (3) relation

deviation, where the motion is visually plausible but does not realize the intended predicate. These cases are removed during filtering and are not used for training. This analysis further motivates our filtering strategy and highlights the importance of preserving instance identity and relation fidelity in generative video augmentation.

**Relation to Temporal Action Localization.** Although TFN also localizes temporal intervals, it differs from standard temporal action localization methods in both supervision and prediction targets. Temporal action localization typically aims to localize action segments from video-level action categories, whereas PVSG requires relation-specific intervals conditioned on subject-object pairs and panoptic entity tubes. Therefore, relation localization in PVSG must jointly consider entity identity, pairwise interaction, and predicate semantics. TFN is designed for this pair-conditioned setting by using entity-pair features and Gaussian masks to guide relation classification.

**Limitations of Generative Augmentation.** RGVA is constrained by the availability of plausible anchor scenes. Relations requiring specific objects, spatial configurations, or fine-grained physical interactions cannot be arbitrarily synthesized without risking semantic inconsistency. As a result, some rare relations receive fewer generated clips than others. Future work may explore stronger controllable video generation models and tighter human-in-the-loop validation to further improve the coverage and reliability of rare-relation augmentation.

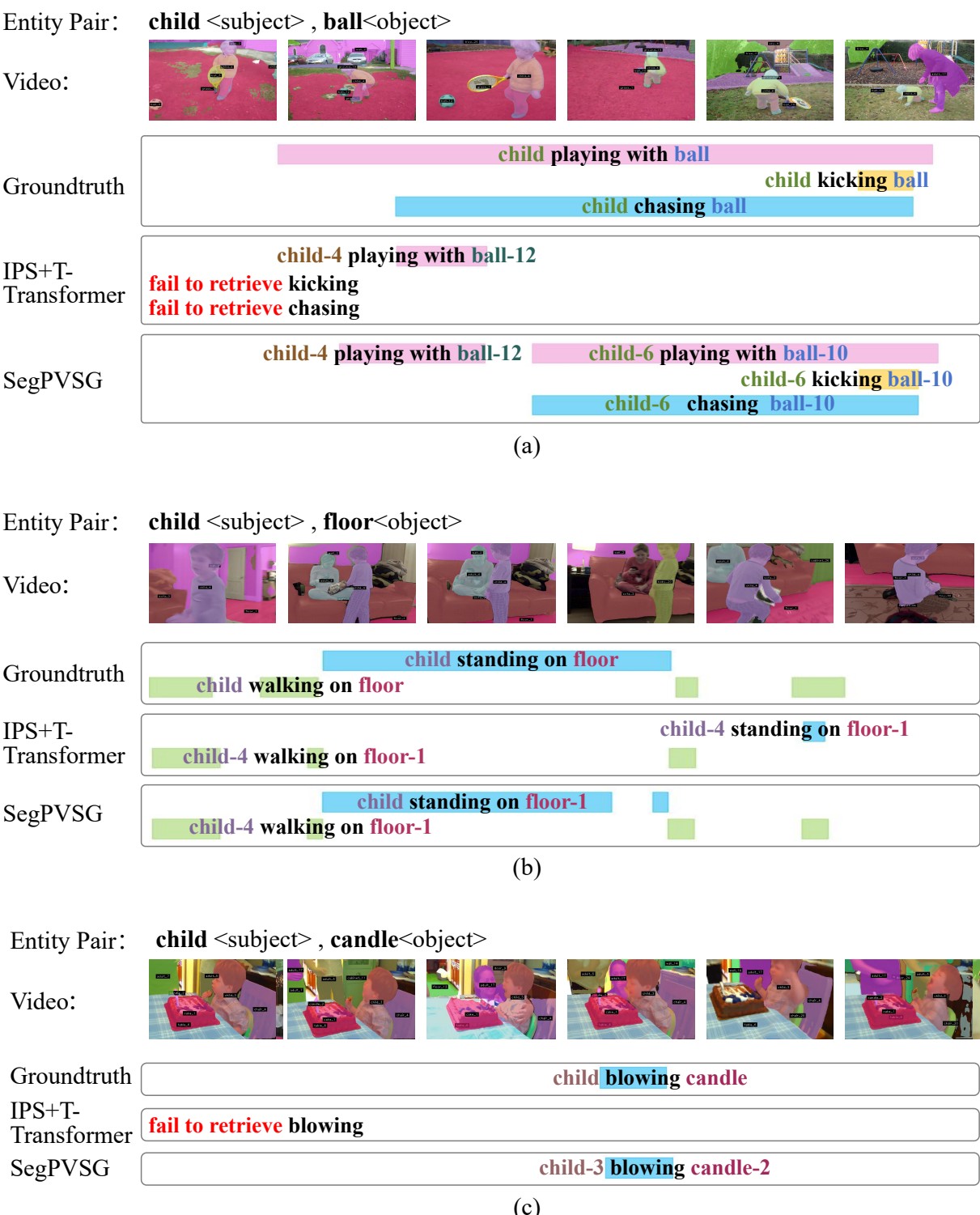

Figure 12. Additional qualitative comparisons between SegPVSG and IPS+T-Transformer (Yang et al., 2023b)

**Role & Goal:**
You are a top-tier Multimodal AI Art Director and Video Choreographer.
Your task is to analyze a scene using dual visual inputs and structured text, propose a single, new, plausible relationship triplet, and then author a descriptive video prompt for generating a "Round-Trip" video clip (where the start and end frames are identical to the raw input).

**Input Information:**
1. **Dual Visual Inputs (CRITICAL):**
   · Image 1 (Raw Image): The primary source of physical truth. Use this to analyze lighting, texture, exact posture, colors, and spatial layout.
   · Image 2 (Mask Visualization): The ID Guide. Objects are colored and labeled with IDs (e.g., '3:person'). You MUST use this to verify which specific object corresponds to the input text list.
2. **Available Objects:** A list of (object_id, class_name) tuples corresponding to the IDs in the Mask Visualization: {objects}
3. **Existing Relations:** The textual description of the scene's ground truth: {relations}
4. **Creative Priority List:** A vocabulary of relations, ordered from rarest/most desired to most common/least desired: {vocabulary}

Your Thinking and Decision-Making Process (A Strict, Step-by-Step Multimodal Process):

**Step 1: Dual-Image Scene Understanding.**
   · Locate IDs in Image 2, then identify their visual appearance in Image 1.
   · Analyze the Raw Image for physical feasibility (posture, obstructions, hand availability).
**Step 2: Generate Theoretical Candidates.**
   · Based on Available Objects and the Creative Priority List, generate theoretically possible (subject, object, relation) combinations.
**Step 3: Apply the Dual-Filter System.**
   · A. Logical Compatibility Filter: Is the candidate compatible with Existing Relations?
   · B. Visual Plausibility Filter (Crucial): Based on the Raw Image, is the action physically possible? Are the subject and object close enough? Are hands free? Does the posture allow it?
**Step 4: Select the Final Choice by Priority.**
   · Choose the highest-priority relation from the Creative Priority List that passes both filters.
**Step 5: Video Action Choreography (The 'Round-Trip' Narrative).**
   · CRITICAL: Video generation models do NOT understand IDs (e.g., "ID:3"). You must TRANSLATE the IDs into rich visual and spatial descriptions based on the Raw Image.
   · **Subject Description:** Describe the subject's appearance (e.g., "the man in the blue shirt", "the brown dog") and position (e.g., "on the left", "in the foreground").
   · **Object Description:** Describe the object's appearance (e.g., "the white ceramic cup", "the red ball") and location.
   · **Narrative Arc:** Describe the "Round-Trip" motion:
        1. Start: The subject starts in the static pose shown in the image.
        2. Action: The subject moves to perform the interaction with the object.
        3. Return: The subject smoothly reverses the motion to return exactly to the initial static pose.
   · **Objective Kinematic Description (De-emotionalized).**
        1. BAN subjective adjectives like "warm", "gentle", "happily", "aggressively".
        2. USE physical descriptors: "slowly extends", "firmly grasps", "tilts head back 30 degrees", "rapidly retracts". Describe ONLY observable physics.
**Step 6: Justify Your Multimodal Decision.**
   · Explain your choice based on visual evidence and priority.

**Output Format:**
Return your proposal in JSON format. The "video_action_prompt" must be a single, fluid natural language paragraph WITHOUT any numerical IDs.

Example if a relationship is found:
{
    "reasoning": "Selected 'Person drinking from Cup'. Visually, the woman (ID 3) in the white dress is seated at a table. The glass (ID 8) is within arm's reach. Her hand is resting on the table, ready to lift it.",
    "new_triple": [3, 8, "drinking from"],
    "video_action_prompt": "The woman in the white dress sitting at the table starts in a static pose with her hand on the table. First, she extends her right forearm forward and opens her fingers to grasp the stem of the clear wine glass. Then, she lifts the glass vertically and tilts it towards her lips to take a sip. Next, she lowers the glass back down, placing it precisely on its original spot on the table. Finally, she retracts her hand back to her lap, returning to her initial static seated posture."
}

Example if NO relationship is plausible:
{
    "reasoning": "No suitable relation found. Hands are occupied and distance is too far.",
    "new_triple": null,
    "video_action_prompt": null
}

*Figure 13.* The full prompt used in RGVA. The MLLM is instructed to generate physically plausible, relation-centric, and round-trip action descriptions conditioned on the anchor frame and instance-level visual grounding.

