# OpenReview forum: "SegPVSG: Panoptic Video Scene Graph Generation via Temporal Focusing and Generative Augmentation"
_ICML.cc/2026/Conference — ICML 2026 regular_

### Official Review · Reviewer_rGFN · 2026-03-07

**Soundness:** 2
**Presentation:** 2
**Significance:** 2
**Originality:** 2
**Overall Recommendation:** 4
**Confidence:** 4

**Summary:**

This paper proposes SegPVSG for panoptic video scene graph generation, targeting temporal sparsity and long-tailed relation distributions. It introduces TempFocusNet (TFN) to localize relation intervals with temporal queries and classify relations using Gaussian mask–guided attention, and adds an MLLM-guided generative augmentation (RGVA) to synthesize and insert rare-relation interaction clips into training videos. Experiments on OpenPVSG and ImageNet-VidVRD show improved performance over prior PVSG baselines.

**Compliance With Llm Reviewing Policy:**

Affirmed.

**Final Justification:**

My concerns have been solved so I would like to increase my score.

**Key Questions For Authors:**

See Weakness

**Limitations:**

Yes

**Strengths And Weaknesses:**

Strength:
1. The TFN module presents a clean decomposition of temporal modeling into interval localization via temporal queries and mask-guided recognition via Temporal Focus Attention. The Gaussian mask mechanism provides an interpretable way to guide temporal focus rather than relying purely on implicit attention dynamics.
2. The RGVA pipeline addresses the severe data scarcity of tail categories by enforcing a closed-loop constraint to synthesize new training samples for rare relations, which ensures background stability.
3. SegPVSG achieves significant gains on the OpenPVSG benchmark, outperforming the leading one-stage model (UNO) by +3.53 mR@20 and +5.90 mR@50 in IPS+T setting.
4. Extensive ablation studies confirm the individual and synergistic effectiveness of the TFN and RGVA modules.

Weaknesses:

1. The benefit of RGVA is not consistent across evaluation settings. Under the VPS setting with vIoU=0.1, RGVA does not show a consistent improvement pattern: by @50, R and mR already trade off against each other, and at @100, RGVA actually degrades performance in both Recall and mean Recall.
2. Distribution Bias Introduced by the Closed-Loop Constraint. The closed-loop design imposes an artificial, reversible temporal structure that may not reflect real-world dynamics. Enforcing identical start and end frames assumes interactions are transient and reversible, potentially biasing the model toward specific motion cues rather than realistic temporal evolution.
3. Issues with Semantic Label Fidelity in RGVA. It remains unclear whether synthesized relations faithfully align with the fine-grained human semantic criteria of PVSG. While visually plausible, generative models risk producing interactions that deviate from the dataset's specific predicate definitions or introduce subtle label drift.
4. Limitation of the Gaussian Temporal Prior. The design implicitly imposes a unimodal and symmetric temporal prior over relation intervals. This design struggles to capture complex real-world dynamics, such as asymmetric motions or interactions occurring across multiple disjoint segments, which cannot be well captured by a single Gaussian-shaped focus distribution.

Minor weakness:
1. Typos：In Fig.1, “glass_4” should be “grass_4”.
2. Ambiguity in Notation Definition. For instance, in line 143, N appears in the formulation M={m1,…,mN}, without a explicitly statement. The definition only becomes clear later in Section 3.1.1. Similarly, the symbol L in line 144 is introduced without a precise definition.

---

> ### Author Rebuttal · Authors · 2026-03-30
>
> We thank the reviewer for the thorough review and valuable feedback, especially the insightful questions regarding our evaluation settings and generation framework.
>
> ## **1. Inconsistent benefit of RGVA under VPS + vIoU=0.1 (W1)**
> We agree that the performance under the VPS + vIoU=0.1 setting deserves clarification. The key issue is that this specific configuration is **considerably noisier** compared to IPS+T or stricter vIoU thresholds.
>
> Under VPS, relation prediction is inherently **affected by upstream segmentation and tracking noise**. Simultaneously, the loose threshold of vIoU=0.1 allows many coarse spatial-temporal overlaps to be counted as hits. Since the evaluation protocol ranks predictions by confidence scores before applying matching rules, this combination makes metrics **highly sensitive to score re-ordering**. While RGVA effectively reallocates probability mass toward rare-relation candidates (which is beneficial in cleaner settings) under "noisy tubelets + loose matching", deeper-ranked predictions (@50/@100) become more vulnerable to noise, leading to less stable Recall behavior.
>
> Importantly, the gains from RGVA **remain consistent and stable under cleaner IPS+T settings or stricter vIoU=0.5 thresholds**. Consequently, we interpret VPS + vIoU=0.1 as a setting that is especially sensitive to upstream noise and score re-ordering, making RGVA’s effect less stable there than under cleaner IPS+T settings or stricter vIoU thresholds.
>
> ## **2. Closed-loop constraint may introduce distribution bias (W2)**
> We agree that the closed-loop constraint is synthetic, but our point is that it is introduced as a controllability mechanism rather than as a claim about real-world temporal evolution. Instead, its purpose is to **create context-consistent, insertable interaction units** that remain visually aligned with the original video while preserving a stable anchor state before and after the inserted event.
>
> 1. **Ensuring Consistency and Suppressing Drift**. Without this constraint, generated clips would be highly susceptible to **background drift and domain mismatch**, which would prevent them from being coherently integrated into the original training stream. This design ensures that the synthetic interaction units maintain **contextual coherence** with the source video.
> 2. **Facilitating Temporal Localization**. A core challenge in PVSG is learning the precise temporal boundaries of interactions. Our closed-loop design enables the model to learn **localized interaction segments within a real-world video context**, rather than from isolated synthetic clips with mismatched temporal boundaries or inconsistent backgrounds.
> 3. **Validation of Generalization Capability**. Notably, **all evaluation is conducted on real-world test videos**, not closed-loop ones. The consistent performance gains observed on real evaluation sets indicate that the model is learning **transferable interaction patterns** rather than overfitting to artificial motion cues.
>
> ## **3. Semantic label fidelity in RGVA (W3)**
> We compare generated clips with real clips using a video-text similarity score and video quality metrics.
>
> 1. **Semantic Fidelity**: We leveraged Qwen-3-VL (2B) for video-text similarity between clips and their corresponding predicate descriptions under the same relation label set.
> 2. **Perceptual Quality**: We employed the DOVER  aesthetic/technical scores, keeping resolution, FPS, and duration consistent.
>
> Results in Table 1,2 show that **our generated videos remain reasonably consistent with target relations while achieving perceptual quality comparable to real-world clips**.
>
> Moreover, real PVSG predicates themselves contain natural variation; **our goal is not to synthesize one canonical motion, but to enrich appearance/motion diversity** while preserving the target relation semantics.
>
> **Table 1: Similarity score statistics (mean±std) for original and generated videos**
> |Video Set|Global|Inter-class|
> |:-|:-:|:-:|
> |Real Videos|0.582±0.093|0.572±0.074|
> |Synthetic Videos|0.549±0.088|0.555±0.054|
>
> **Table 2: Quality assessment scores for original and generated videos**
> |Video Set|Aesthetic score|Technical score|Overall score|
> |:-|:-:|:-:|:-:|
> |Real Videos|93.5±13.0|5.6±1.7|43.9±16.9|
> |Synthetic Videos|93.0±10.7|7.5±1.8|44.9±14.6|
>
> ## **4. Limitation of the Gaussian temporal prior (W4)**
> We agree that a single Gaussian cannot perfectly model all complex patterns. However, TFN **predicts multiple query-specific intervals/masks on the same timeline**, and uses diversity regularization plus score-based merging to cover different candidate spans. This already allows the model to capture repeated or partially disjoint interaction evidence more flexibly than a single prior.
>
> ## **5. Minor issues: typo and notation (MW1,2)**
> Thank you for pointing these out. We will correct "glass_4" → "grass_4" in Fig. 1, and we will define N and L explicitly at their first appearance.

---

> > ### Author Rebuttal · Reviewer_rGFN · 2026-04-03
> >
> > My concerns have been solved so I would like to increase my score.

---

> > > ### Author Response · Authors · 2026-04-03
> > >
> > > Thank you again for your thorough review of our paper and for providing such thoughtful and enlightening feedback. We are truly honored to receive your recognition, which serves as a great encouragement to us. Should you have any further questions, please don’t hesitate to ask; we would be more than happy to address them.

---

### Official Review · Reviewer_pB4V · 2026-03-11

**Soundness:** 4
**Presentation:** 3
**Significance:** 3
**Originality:** 3
**Overall Recommendation:** 4
**Confidence:** 4

**Summary:**

This paper proposes SegPVSG, a framework for Panoptic Video Scene Graph Generation that addresses two challenges: temporal sparsity of relations and long-tailed relation distribution. It introduces TempFocusNet (TFN), which uses learnable temporal queries and Gaussian masks to first localize relation intervals then classify them with focused attention, and Relation-centric Generative Video Augmentation (RGVA), which uses an MLLM and image-to-video model to synthesize closed-loop interaction clips for rare relations and insert them into original training videos. The method achieves improvements of +3.53 mR@20 and +5.9 mR@50 over prior methods on OpenPVSG.

**Compliance With Llm Reviewing Policy:**

Affirmed.

**Final Justification:**

The core idea of the proposed method is novel and effective in VidSGG domain. Moreover, the localization-then-recognition paradigm is well-justified, demonstrating that explicitly predicting relation intervals substantially improves temporal grounding. One concern on this paper is not to address the trade-off between mR@K and R@K of VidSGG domain. However, the authors properly address this issue during the rebuttal. Hence, I am inclined to accept this paper

**Key Questions For Authors:**

- Is the filtering of generated clips in RGVA manual or automated? If manual, what is the annotation cost, and how would this scale to larger datasets?

- What is the computational overhead of RGVA at training time (generation cost, additional training time from longer videos)?

**Limitations:**

yes

**Strengths And Weaknesses:**

## Strengths


- The localization-then-recognition paradigm is well-justified. TFN demonstrates that explicitly predicting relation intervals substantially improves temporal grounding. The idea of narrowing the attention scope before classification is intuitive and effective in VidSGG domain.

- For RGVA, the pipeline of MLLM-guided choreography, closed-loop clip generation, and in-place video insertion is original in this field. The closed-loop constraint is a particularly smart design choice, ensuring the first and last frames match the anchor, it preserves temporal continuity when the clip is injected back into the original video.

- The paper provides extensive ablations covering the number of queries, Gaussian sharpness, diversity weight, individual loss contributions, entity feature representations, and score aggregation strategies.

### Weaknesses

- R@K and mR@K trade-off is not discussed. At vIoU=0.5, SegPVSG achieves 5.45 R@20 versus UNO's 6.23, meaning UNO is better at retrieving the most confident predictions. The mR gains come primarily from tail categories, while head category performance may be comparable or slightly worse. This trade-off deserves explicit discussion, especially for practical applications where top-K precision matters.

- The combination of learnable queries, cross-attention-based interval regression, and Gaussian temporal weighting is well-established in temporal action detection (e.g., ActionFormer, Moment-DETR, GAP). The paper does not sufficiently discuss how TFN's design differs from or improves upon these existing TAD techniques beyond the application domain change.

---

> ### Author Rebuttal · Authors · 2026-03-30
>
> Thank you for the careful reading and for recognizing the motivation of the localization-then-recognition design and the originality of RGVA.
>
> ## **1. Trade-off between R@K and mR@K (W1)**
> We agree that this trade-off warrants a more explicit discussion. **Our method does not simply trade off head category performance to boost tail categories**. As shown in Table 1 below, we provide a fine-grained breakdown of Head, Body, and Tail categories at R/mR@100. The results demonstrate that **SegPVSG achieves consistent performance gains across all three splits, with particularly significant improvements in tail relations**.
>
> Regarding the comparison with UNO, we acknowledge its advantage in R@20, which stems from a one-stage design that favors a concentrated set of highly confident predictions. However, SegPVSG yields a substantially higher mR@20 and maintains stronger performance as $K$ increases, as shown in Table 2. This suggests that **our method produces a more balanced retrieval distribution across categories**.
>
> **Table 1: R/mR@100 results on Head, Body, and Tail splits**
> |Method|Overall|Head (30%)|Body (40%)|Tail (30%)|
> |:-|:-:|:-:|:-:|:-:|
> |Transformer|5.87/4.49|6.56/6.72|3.03/2.46|4.76/4.90|
> |TFN|12.58/12.87|12.85/15.72|14.55/15.57|7.14/6.86|
> |SegPVSG|**13.63/15.91**|**13.12/15.97**|**16.97/18.43**|**9.52/12.75**|
>
> **Table 2: Performance comparison between UNO and our method**
> |Method|R/mR@20|R/mR@50|R/mR@100|
> |:-|:-:|:-:|:-:|
> |UNO ResNet-50|**6.23**/5.60|7.37/6.84|8.65/8.21|
> |TFN|5.14/6.67|9.12/10.15|12.58/12.87|
> |SegPVSG|5.45/**9.13**|**9.54/12.74**|**13.63/15.91**|
>
> ## **2. Relation to temporal action detection (TAD) methods (W2)**
> Thank you for this insightful comment. We agree that TFN shares some surface-level components with prior TAD methods, such as learnable queries and interval regression. However, our contribution is not a direct reuse of these components, but a different way of coupling localization with recognition. Specifically, **TFN is neither a dense per-timestep temporal detector nor a DETR-style set prediction model where each query directly outputs the final semantic prediction**.
>
> Instead, TFN adopts a **localization-conditioned design**: temporal queries first identify candidate interaction intervals from pairwise features, and these intervals are then encoded as Gaussian masks to **serve as interval-aware structural constraints**. Crucially, these masks are not only prediction targets or post-processing refinements; they are **integrated into the Temporal Focus Classifier through row-wise modulation of the attention map**, so that localization directly shapes how temporal evidence is aggregated for relation classification.
>
> This design is particularly suited to PVSG, where relations are temporally fragmented and large portions of a video can be irrelevant to a specific interaction. Rather than recognizing relations from the full temporal extent of an entity pair, TFN first identifies likely interaction spans and then concentrates classification capacity within them while preserving useful global context.
>
> ## **3. Filtering of generated clips and scalability (Q1)**
> Current filtering is performed manually to ensure semantic fidelity, which is critical for rare relations. This process required less than 3 person-hours, **a manageable overhead since augmentation is selectively applied to tail relations rather than the entire dataset**.
>
> To further enhance scalability, we consider that the filtering step can be automated using VLM-based verifiers. We view this **as a promising direction for our future work to extend RGVA to even larger-scale datasets**.
>
> ## **4. Computational overhead of RGVA (Q2)**
> We report both the offline generation cost and training overhead to demonstrate the efficiency of RGVA.
> 1. Offline Preprocessing Cost: RGVA serves as a one-time offline preprocessing step rather than a computationally expensive online training component. As shown in Table 3, **the total API cost for generating the augmented data is approximately $35.2, which is controllable**.
> 2. Training Efficiency: While RGVA extends the average video length, the resulting increase in training time is marginal. Table 4 indicates that **RGVA achieves a significant improvement in mR with only a +8.78% increase in training time per epoch**.
>
> **Table 3: RGVA offline generation cost**
> |Stage|Model|API Calls|Avg. Cost/Call($)|Total Cost($)|Successful API Calls|
> |:-|:-|:-:|:-:|:-:|:-:|
> |Instructions Infer|Seed-1.6|586|0.00404|2.366|586|
> |Video Generate|Seedance-1.5-Pro|570|0.058|32.852|566|
>
> **Table 4: Training overhead with RGVA**
> |Training set|R/mR@100|Training Videos|Total Relations|Avg. Video Length|Time/Epoch|Total Epochs|
> |:-|:-:|:-:|:-:|:-:|:-:|:-:|
> |Origin Set|5.14/6.67|338|4729|75.08s|250.43s|100|
> |Augment Set|5.45/9.13|338|5115|80.56s|272.43s (+8.78%)|100|

---

> > ### Author Rebuttal · Reviewer_pB4V · 2026-04-01
> >
> > Thanks for authors' rebuttal. I will keep my positive score on this paper.

---

> > > ### Author Response · Authors · 2026-04-02
> > >
> > > Thank you again for your thorough review and insightful comments. Your positive feedback is both helpful and encouraging. Should you have any further questions, please don’t hesitate to ask; we would be more than happy to address them.

---

### Official Review · Reviewer_PA9a · 2026-03-12

**Soundness:** 3
**Presentation:** 3
**Significance:** 3
**Originality:** 3
**Overall Recommendation:** 5
**Confidence:** 3

**Summary:**

In this work, the authors propose a model that helps to mitigate two major challenges in the Panoptic Video Scene Graph generation task. The challenges are: 1) Videos are mostly dominated by irrelevant content/background, 2) long tail distribution of relations. To address these issues, the authors propose two components: 1) TempFocusNet, which utilizes learnable queries to search for potential relation intervals, which are further processed using Gaussian temporal masks that aid the model to focus on salient features. 2) Relation-centric generative video augmentation that synthesizes interactive regions to overcome the scarce training data.

**Compliance With Llm Reviewing Policy:**

Affirmed.

**Final Justification:**

My major concerns are addressed. I will retain my positive assessment.

**Key Questions For Authors:**

- What would be the improvement of motion-aware convolution/transformer models when RGCA is used for augmentation?
- Could you provide the computation complexities and parameters used for the proposed model against its baselines?

**Limitations:**

yes

**Strengths And Weaknesses:**

Strengths:
- The paper is well written and easy to follow.
- The experimental results show notable improvement over the latest baselines.
- Experiments are conducted on multiple datasets, and the results are consistent.

Weakness:
- Given that RGVA does synthetic data augmentation, which will enrich the training data, RGVA must also be applied to the best baselines, such as Motion-aware Convolution/Transformer, to clearly differentiate the improvement gained from TFN.

---

> ### Author Rebuttal · Authors · 2026-03-30
>
> Thank you for your valuable feedback and kind recognition of our work. We are grateful that you recognized the motivation, clarity, and consistent gains of our framework.
>
> ## **1. Effect of RGVA on other backbones (W1, Q1)**
> As **the source code for several recent SOTA PVSG methods is not publicly available**, we were unable to integrate RGVA into them within the rebuttal timeframe. We conducted controlled experiments on the most reproducible backbones available.
>
> As shown in Table 1, **RGVA consistently improves multiple backbones**, which suggests that it is a general augmentation strategy. Meanwhile, as shown in Table 2, **TFN itself already substantially outperforms prior motion-aware baselines (MCL)**, and its combination with RGVA yields the best overall performance (see Table 1).
>
> **Table 1: Generalization of RGVA across different backbones**
> |Method|w/ RGVA|R/mR@20|R/mR@50|R/mR@100|
> |-|:-:|:-:|:-:|:-:|
> |Transformer|×|3.88/2.55|5.66/4.12|6.71/5.36|
> |Transformer|√|3.88/3.20|5.45/4.55|7.13/7.19|
> |Convolution|×|3.88/2.81|5.24/3.29|6.18/4.44|
> |Convolution|√|3.88/3.75|6.08/5.71|7.02/6.01|
> |TFN|×|5.14/6.67|9.12/10.15|12.58/12.87|
> |TFN|√|**5.45/9.13**|**9.54/12.74**|**13.63/15.91**|
>
> **Table 2: Comparison between TFN and MCL**
> |Method|R/mR@20|R/mR@50|R/mR@100|
> |-|:-:|:-:|:-:|
> |MCL-Transformer|3.98/2.98|5.97/4.20|7.44/5.15|
> |MCL-Convolution|4.51/3.56|6.08/4.38|7.76/5.86|
> |TFN|**5.14/6.67**|**9.12/10.15**|**12.58/12.87**|
>
> ## **2. Computation complexity and parameter comparison (Q2)**
> To evaluate the parameter count and computational complexity, we conducted inference tests for relation prediction on available baselines under the identical hardware setup (excluding the feature pre-processing stage).
>
> Results in Table 3 shows that TFN introduces only a moderate increase in computational overhead while yielding substantial performance gains. This confirms that **our proposed method achieves a well balance between efficiency and performance**.
>
> **Table3: Complexity and efficiency comparison of different relation modules**
> |Method|R/mR@100|params|FPS|GFLOPS/video|
> |-|:-:|:-:|:-:|:-:|
> |Transformer|6.18/4.44|26.55M|897.74|94.95|
> |Convolution|6.71/5.36|15.76M|802.52|151.86|
> |TFN|12.58/12.87|32.31M|834.12|157.48|

---

> > ### Author Rebuttal · Reviewer_PA9a · 2026-04-03
> >
> > My concerns are partially resolved. I will maintain my original assessment.

---

> > > ### Author Response · Authors · 2026-04-03
> > >
> > > Thank you sincerely for taking the time to thoroughly review our manuscript and for sharing your thoughtful feedback. ​​Your positive feedback is highly valued and encouraging for our research.​​ Should you have any further questions, please don’t hesitate to ask; we would be more than happy to address them.

---

### Official Review · Reviewer_WL96 · 2026-03-12

**Soundness:** 2
**Presentation:** 3
**Significance:** 2
**Originality:** 2
**Overall Recommendation:** 4
**Confidence:** 3

**Summary:**

This paper presents SegPVSG, a temporal-segment-aware framework for Panoptic Video Scene Graph Generation (PVSG). To address temporal sparsity and long-tailed relation distributions , the authors introduce TempFocusNet (TFN), which localizes interaction intervals using Gaussian temporal masks , and Relation-centric Generative Video Augmentation (RGVA), which synthesizes rare-relation training clips via Multimodal LLMs and image-to-video models. The method achieves state-of-the-art results on OpenPVSG and ImageNet-VidVRD , notably improving mean Recall through effective tail-category enhancement.

**Compliance With Llm Reviewing Policy:**

Affirmed.

**Final Justification:**

My concerns have been addressed.

**Key Questions For Authors:**

1.The overall mR@K doesn't explicitly show the improvement on the long-tail issue. Could you provide a detailed breakdown of the Recall metrics across Head, Body, and Tail category splits?
2.Since failure cases are often attributed to tracker ID switches or incomplete masks , can you provide an "Oracle Perception" experiment using Ground-Truth bounding boxes/masks to show the true upper bound of TFN's relation reasoning?
3.How much overhead do the cross-attention and Gaussian mask computations add? Please provide the FPS, Params, and FLOPs compared to the baselines.
4.RGVA strictly relies on existing objects in the anchor frame (e.g., cannot generate "watering" if a "kettle" is missing) . Doesn't this severely limit its scalability for extremely rare tail categories?
5.Why isn't there a comparison with simpler, low-cost long-tail solutions (e.g., loss re-weighting or entity Copy-Paste)? Is the heavy computational cost of video generation truly justified by its performance gain?

**Limitations:**

No. While the authors touch upon general ethical concerns and qualitative failure cases, they fail to adequately address several critical technical limitations inherent to their methodology:
1.	Sensitivity to Base Model Performance: The framework’s efficacy is heavily gated by the performance of the anchor frame mining and the underlying generative models (MLLM and image-to-video). The authors should have discussed how the precision of anchor detection and the "hallucination" rate of the generative model might introduce noisy or physically impossible training signals that could degrade the performance.
2.	Computational and Operational Overhead: The manuscript lacks a rigorous analysis of the additional overhead introduced by the RGVA pipeline, which involves MLLM reasoning, video synthesis, and SAM2-based mask propagation. Without a cost-benefit analysis (e.g., training time or GPU hours vs. mR gain), it is difficult to determine if the marginal performance improvements justify the substantial increase in the complexity of the data preparation pipeline.

**Strengths And Weaknesses:**

** Strengths**

1. Well-Motivated Framework: The paper presents a very clear motivation by directly addressing the two most critical bottlenecks in PVSG: temporal redundancy and the long-tailed distribution of relations. The proposed TFN and RGVA modules are logically derived from these challenges, providing a cohesive and targeted solution.

2. Strong Empirical Results: The experimental validation is comprehensive, showing significant improvements over both one-stage and two-stage baselines. Notably, the gain of +5.9 mR@50 on OpenPVSG highlights the framework's superior ability to retrieve rare and complex relations compared to existing state-of-the-art methods.

**Weaknesses**

1. Using an expensive generative model (RGVA) is a heavy approach. The authors need to compare it against simpler, low-cost baselines (like feature resampling or entity copy-paste) to justify the high computational cost.
2. The quality of generated videos is only supported by a few visual examples without objective metrics (e.g., CLIP-Score). Furthermore, reporting only the overall mR@K  is insufficient; the paper misses standard fine-grained metrics (Head/Body/Tail split recall) to precisely demonstrate the improvements on tail categories.
3. The current end-to-end evaluation is entangled with upstream perception errors, which the authors admit cause many failure cases . The lack of an "Oracle Perception" experiment using Ground-Truth inputs obscures the true relation reasoning upper bound of the proposed TFN module.
4. The paper ignores computational costs. Given the addition of cross-attention and Gaussian masks, the authors must report FPS, parameter counts, and FLOPs to clarify the real-world overhead of this framework.

---

> ### Author Rebuttal · Authors · 2026-03-30
>
> We are grateful for the recognition of our motivation and empirical strength.
> ## **1. Why not compare with simpler long-tail baselines? (W1, Q5)**
> **All models in Table 1 of our paper (including baselines) already incorporate Loss Reweighting** (inverse class frequency) as a standard long-tail strategy. To disentangle RGVA's contribution, we conducted an ablation study. Results in the following Table 1 show that:
> 1. Reweighting is effective, particularly for $mR$ metrics;
> 2. **RGVA alone also improves performance**, confirming it is not redundant;
> 3. **Combining both achieves the best overall performance**, suggesting that RGVA and reweighting are complementary strategies.
>
> **Entity copy-paste is unsuitable for PVSG** as it requires temporally coherent motion, identity preservation, and interval supervision. Naive copy-paste causes severe temporal/label discontinuities, failing as a faithful augmentation.
>
> **Table 1: Ablation study of RGVA and Loss Reweighting**
> |Setting|RGVA|Loss Reweighting|R/mR@50|R/mR@100|
> |:-|:-:|:-:|:-:|:-:|
> |TFN|||9.43/5.12|12.79/8.79|
> |+Reweight||√|9.12/10.15|12.58/12.87|
> |+RGVA|√||9.33/6.15|13.00/11.51|
> |+RGVA +Reweight|√|√|**9.54/12.74**|**13.63/15.91**|
> ## **2. Computational overhead of RGVA (W1, Q5, L2)**
> Due to the page limit, please refer to **Reviewer pB4V, Sec. 4 (Computational overhead of RGVA)** for the detailed response.
> ## **3. Fine-grained long-tail analysis (W2, Q1)**
> We agree that aggregate $mR@K$ may obscure specific gains on rare categories. To clarify, we categorize predicates into Head (top 30%), Body (middle 40%), and Tail (bottom 30%) based on frequency. Table 2 below shows that **SegPVSG achieves consistent performance gains across all splits, particularly in tail categories**. This supports that our method effectively alleviates long-tail bias.
>
> **Table 2: R/mR@100 results on Head, Body, and Tail splits**
> |Method|Overall|Head (30%)|Body (40%)|Tail (30%)|
> |:-|:-:|:-:|:-:|:-:|
> |Transformer|5.87/4.49|6.56/6.72|3.03/2.46|4.76/4.90|
> |TFN|12.58/12.87|12.85/15.72|14.55/15.57|7.14/6.86|
> |SegPVSG|**13.63/15.91**|**13.12/15.97**|**16.97/18.43**|**9.52/12.75**|
> ## **4. Objective quality evaluation of RGVA (W2)**
> Due to the character limit, please refer to **Reviewer rGFN, Sec. 3 (Semantic label fidelity in RGVA)** for the detailed response.
> ## **5. Oracle perception experiment (W3, Q2)**
> To reveal the upper bound of relation reasoning, we replaced predicted masks/tubes with Ground Truth (GT) inputs. The results shown in Table 3 demonstrate:
> 1. The substantial leap suggests **our method is heavily constrained by perception quality**.
> 2. **SegPVSG still maintains a significant lead over TFN** even with GT inputs.
>
> **Table 3: Performance comparison using ground-truth (GT) labels**
> |Setting|R/mR@20|R/mR@50|R/mR@100|
> |:-|:-:|:-:|:-:|
> |TFN|5.14/6.67|9.12/10.15|12.58/12.87|
> |TFN + GT|13.10/15.10|23.38/24.32|32.39/31.75|
> |SegPVSG|5.45/9.13|9.54/12.74|13.63/15.91|
> |SegPVSG + GT|**13.63/15.62**|**26.31/27.80**|**41.09/39.36**|
> ## **6. Computation complexity and parameter comparison (W4, Q3)**
> Due to the character limit, please refer to **Reviewer PA9a, Sec. 2 (Computation complexity and parameter comparison)** for the detailed response.
>
> ## **7. Scalability of RGVA for extremely rare relations (Q4)**
> We acknowledge that RGVA's dependence on existing entities in anchor frames is a limitation. However, **this design is deliberate to ensure scene consistency and generation quality**. Requiring existing objects avoids implausible insertions and keeps the synthesized clip visually coherent with the original video, which is crucial for training interval-aware relation models.
>
> In practice, **this limitation does not prevent gains on rare classes**: the tail-category results exhibited in Table 2 above show substantial improvement. For extremely rare relations, the main challenge is not only low frequency but also the scarcity of physically compatible scenes.
> ## **8. Sensitivity to anchor mining and generator quality (L1)**
> We agree that RGVA depends on the quality of anchor-frame selection and the underlying generative models, and that poor anchors or hallucinated interactions may introduce noisy supervision.
>
> To reduce this risk, RGVA includes multiple safeguards: (1) anchor mining excludes frames inside existing relation intervals, truncated target entities, and low-quality candidates; (2) generation is constrained by scene entities, current context, and the closed-loop requirement to preserve temporal and visual consistency; and (3) generated clips are further filtered based on target-pair consistency, object continuity, and relation validity. In addition, RGVA is only applied to rare relations, so the scope of potential noise is limited.
>
> Empirically, the retained clips are beneficial. **RGVA consistently improves different backbones**, and our objective evaluations show that **generated clips remain close to real clips in both semantic alignment and perceptual quality**.

---

> > ### Author Rebuttal · Reviewer_WL96 · 2026-04-03
> >
> > The authors addressed some concerns using Head/Body/Tail metrics and Oracle experiments. However, the practical overhead of the generative pipeline remains insufficiently justified. Without explicit efficiency data (FPS/FLOPs) to compare against low-cost baselines, the trade-off between high complexity and performance gains isn't fully clear. Thus, concerns are partially resolved.

---

> > > ### Author Response · Authors · 2026-04-03
> > >
> > > We appreciate the reviewer's recognition of our improvements. Regarding practical overhead, we agree that explicit efficiency data is essential. Here, we provide detailed FPS/FLOPs comparisons for our model architecture, and clarify the overhead compared to low-cost baselines (which we previously addressed in our responses to Reviewer PA9a and Reviewer pB4V). Our results, summarized as follows, demonstrate that **RGVA incurs controllable overhead, ensuring that the performance gains justify the complexity**.
> > >
> > > 1. **RGVA overhead is a manageable, one-time offline cost**. It is worth clarifying that **RGVA serves entirely as a one-time offline preprocessing step, rather than a computationally expensive online training component**. As detailed in Table 1, the total API cost for generating the augmented data is approximately $35.2. This demonstrates that the monetary cost of the generative pipeline is highly affordable and strictly bounded.
> > >
> > > 2. **The impact of RGVA on training time is marginal**. While the RGVA-augmented data slightly extends the average video length, **the resulting increase in training time is minimal**. As shown in Table 2, incorporating RGVA achieves a significant improvement in mR with only an 8.78% increase in training time per epoch.
> > >
> > > 3. **RGVA and low-cost baselines introduce zero inference overhead and are highly complementary**. Regarding the comparison with low-cost long-tail baselines (e.g., Loss Reweighting), it is important to clarify their computational footprint. **Loss Reweighting introduces no additional time cost during either training or inference**. Similarly, our **RGVA introduces zero additional computational overhead during inference** (as it acts as data augmentation). To address your concern about whether RGVA is necessary given such low-cost alternatives, we evaluated their independent and combined performance impacts (Table 3). The results indicate that:
> > >     - Using either Loss Reweighting alone or RGVA alone improves performance, confirming that **RGVA provides independent benefits**.
> > >     - Combining both yields the best overall performance, demonstrating that **RGVA and reweighting are complementary rather than mutually exclusive**.
> > >
> > > 4. **Explicit efficiency data (FPS/FLOPs) confirms a favorable complexity-performance trade-off**. To directly address your concern regarding inference complexity, we evaluated the computational complexity and parameter count of our relation module (TFN) against other model architectures. We conducted inference tests under an identical hardware setup (excluding the feature pre-processing stage). As shown in Table 4, while TFN introduces a moderate increase in GFLOPs and parameters compared to the Transformer or Convolution baselines, **it maintains a highly competitive real-time inference speed (834.12 FPS) and yields substantial performance gains (e.g., mR@100 increases from 5.36 to 12.87)**. This explicit efficiency data confirms that our proposed method achieves an optimal trade-off between computational complexity and performance gains.
> > >
> > >
> > > **Table 1: RGVA offline generation cost**
> > > |Stage|Model|API Calls|Avg. Cost/Call($)|Total Cost($)|Successful API Calls|
> > > |:-|:-|:-:|:-:|:-:|:-:|
> > > |Instructions Infer|Seed-1.6|586|0.00404|2.366|586|
> > > |Video Generate|Seedance-1.5-Pro|570|0.058|32.852|566|
> > >
> > > **Table 2: Training overhead with RGVA**
> > > |Training set|R/mR@100|Training Videos|Total Relations|Avg. Video Length|Time/Epoch|Total Epochs|
> > > |:-|:-:|:-:|:-:|:-:|:-:|:-:|
> > > |Origin Set|5.14/6.67|338|4729|75.08s|250.43s|100|
> > > |Augment Set|5.45/9.13|338|5115|80.56s|272.43s (+8.78%)|100|
> > >
> > > **Table 3: Ablation study of RGVA and Loss Reweighting**
> > > |Setting|RGVA|Loss Reweighting|R/mR@50|R/mR@100|
> > > |:-|:-:|:-:|:-:|:-:|
> > > |TFN|||9.43/5.12|12.79/8.79|
> > > |+Reweight||√|9.12/10.15|12.58/12.87|
> > > |+RGVA|√||9.33/6.15|13.00/11.51|
> > > |+RGVA +Reweight|√|√|**9.54/12.74**|**13.63/15.91**|
> > >
> > > **Table4: Complexity and efficiency comparison of different relation modules**
> > > |Method|R/mR@100|params|FPS|GFLOPS/video|
> > > |-|:-:|:-:|:-:|:-:|
> > > |Transformer|6.18/4.44|26.55M|897.74|94.95|
> > > |Convolution|6.71/5.36|15.76M|802.52|151.86|
> > > |TFN|12.58/12.87|32.31M|834.12|157.48|

---

### Decision · Program_Chairs · 2026-04-30

**Decision:**

Accept (regular)

**Comment:**

The paper proposes SegPVSG, a temporal-segment-aware framework for Panoptic Video Scene Graph Generation (PVSG) that addresses temporal sparsity and long-tailed relation distributions via a temporal focusing module (TFN) and a generative augmentation pipeline (RGVA).

The reviewers acknowledged several strengths of the work: (i) a clear and well-motivated formulation; (ii) a principled localization-then-recognition design (TFN) that improves temporal grounding; (iii) a novel generative augmentation pipeline (RGVA) with a closed-loop design to synthesize rare-relation video clips while preserving temporal consistency; (iv) strong and consistent empirical improvements across multiple datasets and settings, with comprehensive experimental validation. Nevertheless, several key concerns were also raised in the initial reviews: (1) computational cost and practicality, particularly the heavy reliance on generative augmentation and lack of initial efficiency analysis; (2) evaluation completeness, including missing fine-grained long-tail breakdowns, oracle perception analysis, and insufficient comparison with simpler baselines; (3) stability and generalization of RGVA, including potential distribution bias from the closed-loop design and inconsistent gains under certain settings; and (4) methodological clarity, such as the relation to prior temporal modeling approaches and the trade-off between Recall and mean Recall.

During rebuttal discussion, the authors addressed these concerns with additional experiments and clarifications: 1) Head/Body/Tail breakdowns and oracle perception experiments to better validate long-tail improvements and reasoning capability; 2) efficiency metrics (FPS, FLOPs, parameters) and training overhead, clarifying that RGVA is primarily an offline augmentation step with limited runtime impact; 3) complementary with standard long-tail techniques and generalization across backbones; and 4) additional analysis on RGVA quality, distribution bias, and robustness. These responses resolved most concerns, though some reviewers noted that the cost–benefit trade-off and stability under certain evaluation settings could be further clarified. After the rebuttal, all reviewers maintained or strengthened their positive assessments, reaching a consensus.

The AC agrees with the reviewers that the paper presents a well-motivated and empirically strong approach to PVSG, with both methodological and practical contributions. While the computational complexity and some design assumptions warrant further discussion, the rebuttal has sufficiently addressed the main concerns. Therefore, the AC recommends accept, and encourages the authors to further clarify efficiency trade-offs, simplify presentation where possible, and expand analysis of generative augmentation robustness in the final manuscript.